# OMI-observed HCHO in Shanghai, China during 2010-2019 and ozone sensitivity inferred by improved HCHO/NO₂ ratio

Danran Li [1], Shanshan Wang [1,2], Ruibin Xue [1], Jian Zhu [1], Sanbao Zhang[1], Zhibin Sun[1], and Bin Zhou [1,2,3]

[1]Shanghai Key Laboratory of Atmospheric Particle Pollution and Prevention (LAP[3]), Department of Environmental Science and Engineering, Fudan University, Shanghai, China
[2]Institute of Eco-Chongming (IEC), No. 20 Cuiniao Road, Shanghai 202162, China
[3]Institute of Atmospheric Sciences, Fudan University, Shanghai, 200433, China

*Correspondence to*: Shanshan Wang (shanshanwang@fudan.edu.cn)

**Abstract.** In recent years, satellite remote sensing has been increasingly used in the long-term observation of ozone ($O_3$) precursors and its formation regime. In this work, formaldehyde (HCHO) data from Ozone Monitoring Instrument (OMI) were used to analyse the temporal and spatial distribution of HCHO vertical column densities (VCDs) in Shanghai from 2010 to 2019. HCHO VCDs exhibited the highest value in summer and the lowest in winter, the high-VCD concentrated in western Shanghai. Temperature largely influence HCHO by affecting the biogenic emissions and photochemical reactions, 15 and industry was the major anthropogenic source. The satellite observed formaldehyde to nitrogen dioxide ratio ($FNR_{SAT}$) reflects that the $O_3$ formation regime had significant seasonal characteristics and gradually manifested as transitional ozone formation regime dominated in Shanghai. The uneven distribution in space was mainly reflected as the higher $FNR_{SAT}$ and surface $O_3$ concentration in suburban area. To compensate the shortcoming of $FNR_{SAT}$ that it can only characterize $O_3$ formation around satellite overpass time, correction of $FNR_{SAT}$ was implemented with hourly surface FNR and $O_3$ data. 20 After correction, $O_3$ formation regime showed the trend moving towards VOC-limited in both time and space, and regime indicated by $FNR_{SAT}$ can better reflect $O_3$ formation for a day. This study can help us better understand HCHO characteristics and $O_3$ formation regime in Shanghai, and also provide a method to improve $FNR_{SAT}$ for characterizing $O_3$ formation in a day, which will be significant for developing $O_3$ prevention and control strategies.

## 1 Introduction

Formaldehyde (HCHO) is an important trace gas in atmosphere. It has the irritating effect on human eyes, skin, and respiratory mucosa, and can also cause cancer in high concentration (Zhu et al., 2017a; Liu et al., 2018a). Atmospheric HCHO is an intermediate product of almost all volatile organic compounds (VOCs) oxidation, it can be therefore indicative of the overall VOCs level (Chan et al., 2019). HCHO can also be emitted through anthropogenic sources, biogenic sources, and biomass combustion. Anthropogenic sources like transportation, power, industry, and residential etc. increase the 30 amount of HCHO by emitting VOCs into the atmosphere (Wang et al., 2017). In addition, biogenic volatile organic

compounds (BVOCs) are also important sources of HCHO. Isoprene emitted by plant can be oxidized to generate HCHO, which causes the concentration of HCHO in some lush vegetation areas to be largely affected by emission of BVOCs (Millet et al., 2008). The removal of HCHO is mainly through photolysis, reaction with OH radicals and the deposition (Ling et al., 2016; Xing et al., 2020).

Satellite remote sensing can achieve large-scale observation of atmospheric pollutant gases including HCHO, which has been widely used in recent years. Sensors currently available for HCHO observation include the Global Ozone Monitoring Experiment (GOME) on ERS-2 (Burrows et al., 1999; Martin et al., 2004b), Scanning Imaging Absorption Spectrometer for Atmospheric Chartography (SCIAMACHY) on ENVISAT (Bovensmann et al., 1999; Stavrakou et al., 2009), Ozone Monitoring Instrument (OMI) on Aura (Levelt et al., 2006; Zhu et al., 2017b), GOME-2 A, B, and C on METOP as the
successor of GOME (Callies et al., 2000; De Smedt et al., 2012), Ozone Mapping and Profiler Suite (OMPS) on Suomi-NPP (Su et al., 2019), and Tropospheric Monitoring Instrument (TROPOMI) on Sentinel-5P (Veefkind et al., 2012; Vigouroux et al., 2020). OMI can provide daily data of HCHO with higher spatial resolution (13 km×24 km). As a new generation of sensors, TROPOMI was launched in 2017, it has better spatial resolution (7 km×7 km) but lacks long-term observation so far. It would be an advantageous tool of satellite remote sensing to achieve more detailed analysis in the future (Veefkind et al.,
45    2012).

Previous studies reported satellite observed long-term and large-scale distribution and variation of HCHO in China and all over the world (Millet et al., 2008; Zhu et al., 2017a; Liu et al., 2020). Twelve years observation of multi-satellite (OMI, GOME-2, SCIAMACHY) showed that the trend of HCHO vertical column densities (VCDs) over eastern China is consistent with that of anthropogenic VOCs (Shen et al., 2019). Based on 10 years observation of OMI HCHO, Liu et al.
(2018a) indicated that high HCHO VCDs in tropical forests region are greatly affected by biomass burning and meteorological factors including temperature and precipitation. In addition, the ground-based remote sensing can also be available for HCHO observation, such as the multi axis differential optical absorption spectroscopy (MAX-DOAS) measurement. The vertical distribution of HCHO derived from MAX-DOAS measurement was characterized by the higher HCHO concentrated near the surface (Lee et al., 2015; Chan et al., 2019). By employing the box model, Li et al. (2014)
found that isoprene oxidation initiated by OH radicals have a great contribution to the HCHO formation in semi-rural region of the Pearl River Delta (PRD) in China.

HCHO participates in the complex photochemical reaction of $NO_x$ ($NO_x$= NO + $NO_2$) and directly affects the production of $O_3$ in troposphere. Due to the short lifetime of HCHO and $NO_2$, their spatial distributions were greatly affected by local emission of VOCs and $NO_x$, which received widespread attention as precursors of tropospheric $O_3$ (Zaveri et al., 2003; Chan
et al., 2019). Consequently, HCHO and $NO_2$ can be assumed as indicators of VOCs and $NO_x$, and the ratio of formaldehyde to nitrogen dioxide (HCHO/$NO_2$, FNR) can be an indicator to analyse the $O_3$ formation regime (Sillman, 1995; Martin et al., 2004a; Schroeder et al., 2017). For instance, by using FNR from long-term OMI HCHO and $NO_2$ data, $O_3$ sensitivity of the United States was evaluated. $O_3$ formation regime can be designated as VOC-limited for FNR < 1, $NO_x$-limited for FNR > 2, and transition for 1 < FNR < 2, which serves as the transitional regime between VOC-limited and $NO_x$-limited regimes,

indicating the production of $O_3$ can be changed by both VOC and $NO_x$ (Duncan et al., 2010). In view of China, OMI products over three representative regions (North China Plain (NCP), the Yangtze River Delta (YRD) and the PRD) were investigated, revealing that the $O_3$ formation regime varied in both time and space, and the contribution of emission sectors to precursors changed with the type of regimes (Jin and Holloway, 2015). During special events such as Asia-Pacific Economic Cooperation in 2014 and Grand Military Parade in 2015, FNR in Beijing had become higher compared with

previous periods, and the $O_3$ formation regime shifted toward $NO_x$-limited regime with control strategies (Liu et al., 2016). In this study, OMI satellite data were used to investigate the temporal and spatial distribution characteristics of atmospheric HCHO in Shanghai from 2010 to 2019, combined with meteorological data and emission inventories to analyse the influencing factors. FNR calculated by satellite HCHO and $NO_2$ were applied to capture variation of the $O_3$ formation regime in Shanghai over the past decade. Considering that satellite data only reflect the column density of trace gas around overpass

time, hourly surface FNR and $O_3$ concentration increment were proposed to correct the satellite FNR, so that it can better indicate $O_3$ formation in the daytime.

## 2 Data and Methods

### 2.1. Satellite data

OMI on Aura orbits the earth in about 98 minutes, which can achieve full coverage of the earth in one day. It overpasses at

13:45 local time (LT) each day. The scanning width is 2600 km, and is divided into 60 rows. The sensor contains 3 channels, including UV-1, UV-2, and VIS, with a wavelength coverage of 264-504 nm. This band allows to observe a variety of trace gases, e.g., HCHO, $NO_2$, and $SO_2$ (Zhang et al., 2019). The retrieval algorithm of this product is based on nonlinear least-squares fitting which get slant column density (SCD) as the result. Then SCD can be converted to VCD through Air Mass Factors (AMF). The Level-2 OMI HCHO product OMHCHO Version-3 is used in this study (https://disc.gsfc.nasa.gov).

Since atmospheric HCHO is mainly distributed in the troposphere, the total VCD can be regarded as the tropospheric VCD of HCHO (Duncan et al., 2010). The Level-2 OMI $NO_2$ product OMNO2.003 Version-4 is adopted as tropospheric $NO_2$ VCD in this study (https://disc.gsfc.nasa.gov).

### 2.2. Methodology

In this study, Shanghai and surrounding areas were gridded into to a spatial resolution of $0.01°\times0.01°$. Then pixel falling

within a 24 km radius of the grid center were averaged, and further was assigned to that grid (Fioletov et al., 2011; McLinden et al., 2012; Zhu et al., 2014). In order to remove data with poor quality as much as possible, HCHO data with cloud fraction $\leq 30\%$, solar zenith angle $\leq 70°$, and Main Data Quality Flag = 0 were selected in this study. In addition, the quality of pixel data with large size is poor, so 5 marginal pixels on each side were abandoned, and only pixel data within 6~55 were selected (Zhu et al., 2017a; Xue et al., 2020). Because OMI has experienced row anomaly since 2007, Xtrack flag

=   0   was   required   to   eliminate   the   influence   of   poor   quality   data   affected   by   row   anomaly

(http://projects.knmi.nl/omi/research/product/rowanomaly-background). As HCHO satellite data have large error, fitting root mean square (RMS) ≤ 0.003 was limited to remove most outliers (Souri et al., 2017). The selection of $NO_2$ satellite data was basically the same as that of HCHO, but without fitting RMS parameter filtering, and cloud radiance fraction ≤30% was required (Krotkov et al., 2016; Xue et al., 2020). Moreover, the linear regressions of monthly deseasonalized zonal mean HCHO VCDs with 0.5° latitude steps over remote pacific region (29° - 33 °N, 160° - 140 °W) indicated that OMI HCHO product used in this study do not show the obvious drift (Zhu et al., 2017b).

### 2.3. Auxiliary data

The meteorological data, including monthly temperature, sunshine hours, precipitation, and relative humidity are acquired from the National Bureau of Statistics of China (http://www.stats.gov.cn/). Hourly temperature data at Shanghai Hongqiao INTL Site (31.20°N, 121.34°E) come from the National Climatic Data Center (NCDC, https://www.ncdc.noaa.gov/). The anthropogenic sources of HCHO in Shanghai are calculated based on the Multi-resolution Emission Inventory for China (MEIC, http://www.meicmodel.org/). Surface HCHO and $NO_2$ concentrations were measured by long-path differential optical absorption spectroscopy (LP-DOAS) at the Jiangwan campus of Fudan University in Shanghai (31.34°N, 121.52°E). The $O_3$ data of Qingpu Dianshan Lake Site (31.09°N, 120.98°E) and Hongkou Site (31.30°N, 121.47°E) in Shanghai are obtained from the Shanghai Environmental Monitoring Center (http://www.semc.com.cn/aqi/Home/Index).

## 3. Results and Discussion

### 3.1. The temporal and spatial variation of HCHO

HCHO VCDs for annual, monthly and seasonal variations in Shanghai from 2010 to 2019 are shown in Fig. 1. HCHO VCDs decreased from the highest value of $12.78\times10^{15}$ molec·$cm^{-2}$ in 2010 to the lowest value of $10.37\times10^{15}$ molec·$cm^{-2}$ in 2012, then rebounded from 2012 to 2014, and fluctuated slightly in the following years (Fig. 1a). The column value and variation are similar to previous study in the YRD, China (Zhang et al., 2019). Before 2018, the high-VCD concentrated in June to August for about $15\times10^{15}$ to $20\times10^{15}$ molec·$cm^{-2}$, the low-VCD appeared in January to February and November to December for about $3\times10^{15}$ to $9\times10^{15}$ molec·$cm^{-2}$, and VCDs were comparable in remaining months (Fig. 1b). In addition, the amplitude of monthly HCHO VCDs was relatively smaller in 2018 and 2019, and mainly concentrated from $6\times10^{15}$ to $15\times10^{15}$ molec·$cm^{-2}$. The HCHO VCDs varied with the season, the maxima and minima corresponding to the respective summer (June, July and August) and winter (December, January and February), whereas moderate levels in spring (March, April and May) and autumn (September, October and November) (Fig. 1c). Previous MAX-DOAS and OMI observations also exhibited the same seasonal patterns of HCHO in the YRD, China (Jin and Holloway, 2015; Chan et al., 2019). High temperature and abundant radiation are conducive to the plant growth to produce BVOCs and the photochemical reaction of VOCs, which boost the HCHO formation in summer (Sharkey and Loreto, 1993; Duncan et al., 2009; Narumi et al., 2009). Thus, HCHO VCDs would be relatively low in winter under the opposite weather conditions.

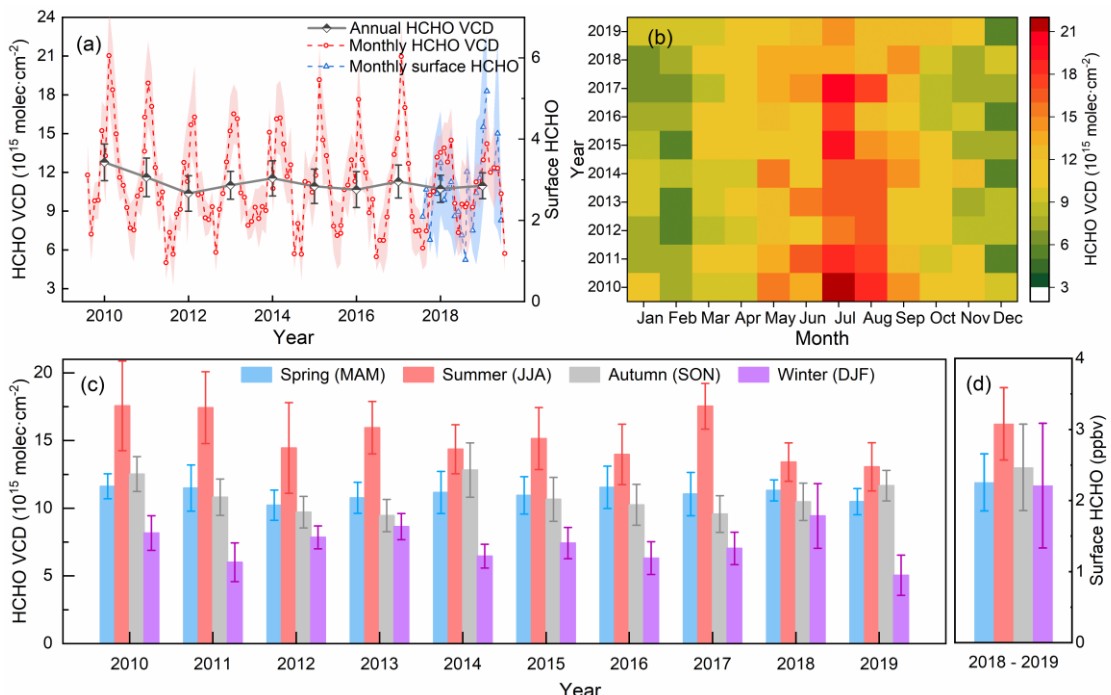

**Figure 1. OMI and LP-DOAS observed time series of HCHO in Shanghai. (a) to (c) reflect the annual, monthly and seasonal variations of HCHO VCD during 2010-2019, (d) reflects the seasonal variation of surface HCHO during 2018-2019.**

The HCHO VCDs within 10 km of LP-DOAS measurement site were averaged to compare with the surface HCHO between 13:00 and 14:00 (Fig. S1). It shows that HCHO VCDs and surface HCHO concentrations are not consistent very well, as HCHO is not completely concentrated near the ground, but has a high concentration at higher altitudes (Chan et al., 2019; Wang et al., 2019). Spatial heterogeneity of surface HCHO in horizontal can also impact the consistency of the comparison. However, the surface HCHO observed by LP-DOAS shows the same seasonal characteristics as HCHO VCDs (Fig. 1d).

The spatial distribution of 10-year averaged HCHO VCD was given in Fig. 2a. In general, HCHO VCDs in eastern coastal area were relatively low, with the level of about $10 \times 10^{15}$ molec·cm$^{-2}$. While those in western regions adjacent to other provinces were relatively higher, about $13 \times 10^{15}$ molec·cm$^{-2}$. Noted that area with the highest HCHO VCD in Shanghai not appeared in the city center (marked by the red box) but in the relatively remote Qingpu (QP) district, followed by Songjiang (SJ) and Jinshan (JS) district. Compared to the anthropogenic Non-methane volatile organic compounds (NMVOCs)

emissions, the distribution of HCHO VCDs does not show the same spatial pattern (An et al., 2021). The high HCHO in western Shanghai were frequently observed and may be explained by the transport of air masses containing high concentrations of reactive VOCs sometimes from Zhejiang and Jiangsu provinces and the significant contribution of local biogenic isoprene to HCHO (Su et al., 2019; Zhang et al., 2020; Zhang et al., 2021).

    Figure 2b shows the difference of HCHO VCDs between 2019 and 2010. It suggests that except for the eastern and southern

coastal areas, as well as the eastern area of Chongming Island, HCHO VCDs in Shanghai showed an overall downward trend

during the past 10 years, with western regions experiencing the largest decline. Figure 2c to Fig. 2f display the spatial distribution of HCHO VCDs in different seasons. HCHO VCD in summer was basically above $12\times10^{15}$ molec·cm$^{-2}$. In winter, the value was around $7\times10^{15}$ molec·cm$^{-2}$ for most regions except for Qingpu district. While in spring and autumn, it was in the moderate level of about $10\times10^{15}$ molec·cm$^{-2}$. The spatial distribution of HCHO VCDs in different seasons was similar to the 10-year averaged characteristics of high-value in the west and low-value in the east.

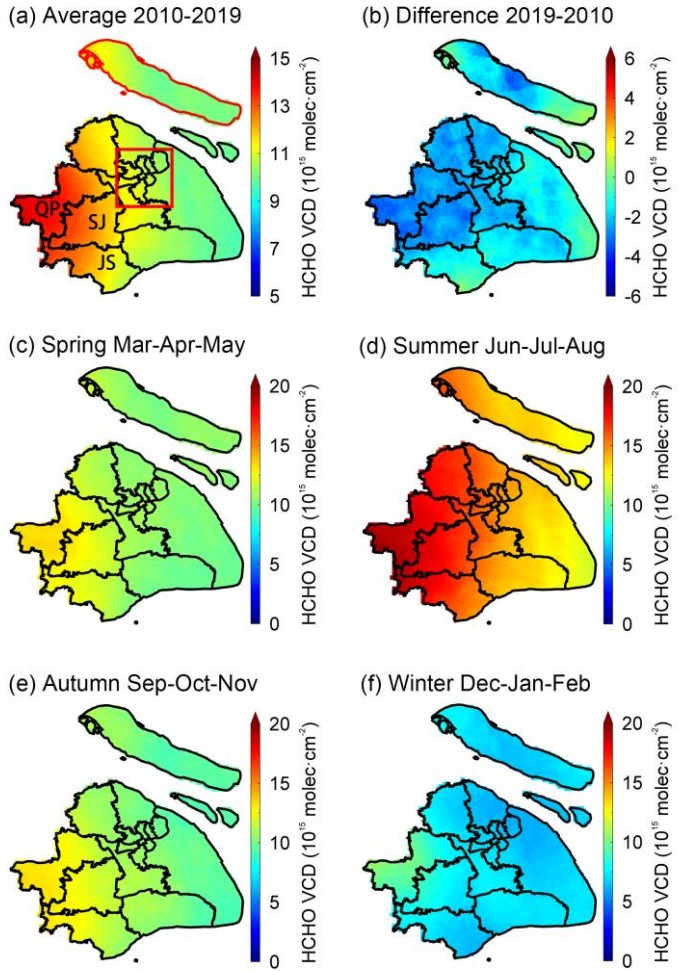

**Figure 2. Spatial distribution of HCHO VCDs in Shanghai: (a) average HCHO VCD for 10 years, (b) the difference of HCHO VCDs between 2019 and 2010 (2019 minus 2010), and (c) to (f) for different seasons. In Fig. 2a the city center is marked with red box, Chongming Island is displayed with red boundary, QP, SJ, and JS refer to Qingpu, Songjiang, and Jinshan district.**

### 3.2. Influencing factors

The relationship between monthly HCHO VCDs and meteorological variables including temperature, sunshine hours, precipitation, and relative humidity were analysed via the linear regression. The stepwise regression results show that only temperature correlated with HCHO VCDs significantly among these four meteorological factors. The linear regression of

temperature contribution and HCHO VCD was shown in Fig. 3. The temperature contribution has a good correlation with the observed HCHO VCD ($R^2 = 0.73$), which means that temperature can explain about 73% of the variation of HCHO VCD. The remaining part that cannot be explained by temperature appears in the form of residual, which is considered as the influence of other changing factors such as anthropogenic emissions (Li et al., 2019). The residual in summer in some years would be particularly large, which indicates that in addition to temperature, there are other factors affecting HCHO VCD significantly.

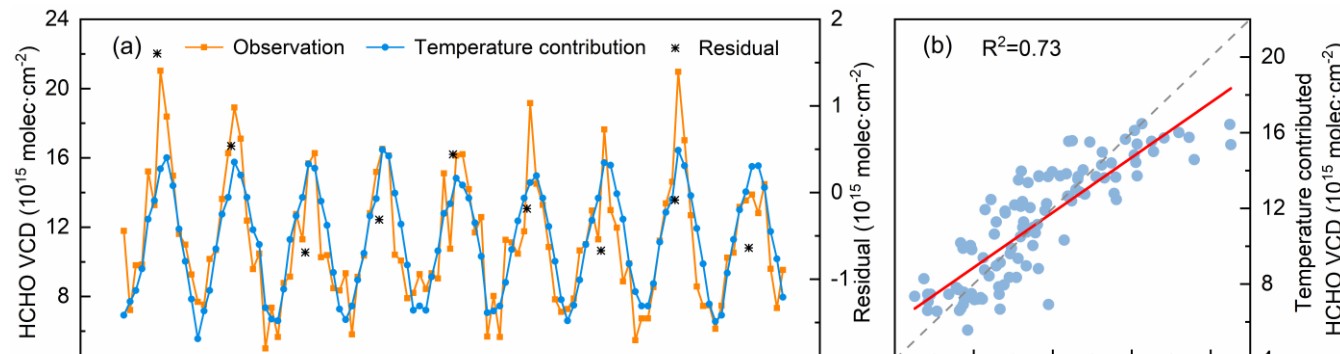

**Figure 3. Monthly HCHO VCDs and the temperature contribution in Shanghai during 2010-2018. (a) reflects the temporal variations and (b) illustrates the correlation analysis. The black points represent the annual average residuals.**

The precipitation and relative humidity reached peak in June, while the sunshine hours reached the dip, and HCHO VCD declined with the temperature rose (Fig. S2). Shanghai has a subtropical monsoon climate with rain and heat in the same period. Abundant precipitation largely favoured the wet deposition of HCHO and offset the impact of rising temperature, resulting in a small decrease in HCHO VCDs in June (Pang et al., 2009). The result of Fisher's exact test also illustrates that when the relative humidity changes remarkably, the variation of HCHO VCDs in summer would be significantly affected ($P < 0.05$).

To explore the impacts of anthropogenic sources on HCHO abundance, primary emissions and secondary productions of HCHO from anthropogenic sources were estimated. NMVOCs emissions from MEIC v.1.3 grid inventories for the year of 2010, 2012, 2014, and 2016 were mapped to SAPRC07 mechanism species. Primary HCHO was directly obtained from the inventory, secondary HCHO production was calculated based on 1-day HCHO yields of several NMVOCs under high-$NO_x$ condition (Table S1) (Shen et al., 2019).

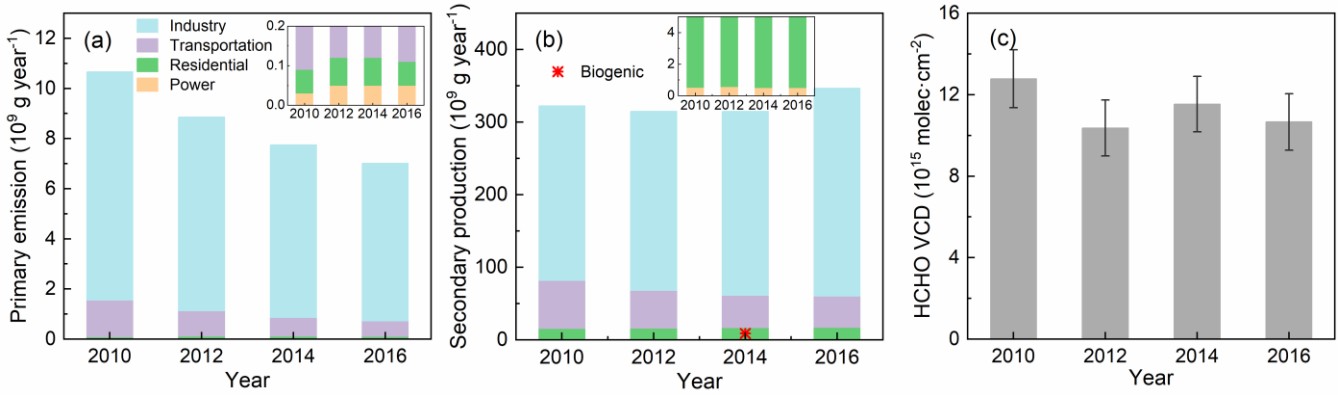

**Figure 4. Primary and estimated secondary production of HCHO from anthropogenic NMVOCs, biogenic contribution of HCHO in 2014 was marked as red cross in Figure 4(b).**


In Fig. 4a and 4b, for the total of primary emissions and secondary productions, the contribution of industrial sector was much higher than other sectors, accounting for about 79.3% of the total, followed by transportation sector, accounting for about 15.7%. Residential and power sectors were far lower, the proportions over these years were basically less than 5%. As major sectors, the contribution of industrial increased 17.2%, while the contribution of transportation decreased about 35.2%.

The primary emission of HCHO keeps decreasing (about 34.2% compared to 2010), while secondary produced HCHO did not change significantly. Furthermore, the changes and proportional relationships between primary emission and secondary production of HCHO for these sectors are different, which means the VOCs source profile of different emission sectors would affect the amount of secondary HCHO production. It should be noted that the estimation here is the potential of secondary HCHO production from anthropogenic NMVOCs, and does not represent the actually produced in the atmosphere.

In addition, HCHO produced by biogenic sources in 2014 was also calculated, and the BVOCs emissions were referred from Liu et al. (Liu et al., 2018b; 2018c). HCHO yield from BVOCs emission was about $9.07 \times 10^9$ g, which is much smaller compared with anthropogenic contribution, indicating that the anthropogenic is the main contributor of secondary production of HCHO in Shanghai (Shen et al., 2019; Fan et al., 2021). The spatial distribution of surface land types in Shanghai (Fig. S3) also suggest the less emission and contribution of biogenic sources in urbanized and densely populated area compared to

forested regions.

Fig. 4c shows that the annual trend of HCHO VCDs is not synchronized with that of primary or secondary HCHO. It means that, besides the combined effect of the primary and secondary source, the changes of HCHO VCDs should also be affected complexly by various factors. In addition, it should be noted that HCHO yield from VOCs is proportional to $NO_x$ condition (Palmer et al., 2006; Marais et al., 2012; Miller et al., 2017). With context of the continuous $NO_x$ decreases in Shanghai, the

estimation using a fixed HCHO yield may overestimate secondary HCHO production in later years (Xue et al., 2020). However, the $NO_x$ concentration in Shanghai (basically 30-60 ppbv in urban) is still much higher than the defined high $NO_x$

condition (1 ppbv) in previous studies (Gao et al., 2017). Therefore, in such a high $NO_x$ condition, the effect of $NO_x$ decreases on HCHO yield needs to be further studied.

### 3.3. FNR and $O_3$ formation regime

As the important precursors of $O_3$, HCHO and $NO_2$ can be served as indicators for VOCs and $NO_x$. On this basis, the $HCHO/NO_2$ ratio from satellite observation ($FNR_{SAT}$) can be employed to identify the $O_3$ formation regime. The variations of monthly averaged VCDs of HCHO, $NO_2$, and $FNR_{SAT}$ in Shanghai over the past 10 years are given in Fig. 5. $NO_2$ VCDs were featured by the highest in winter and the lowest in summer, which was opposite to HCHO and fluctuated more fiercely. The peaks were mainly on account of its longer lifetime in winter (Zhang et al., 2007). While in summer, the adequate

sunlight and precipitation accelerated the photochemical removal and wet deposition of $NO_2$, resulting in the dip (Wang et al., 2018; Xue et al., 2020). $FNR_{SAT}$ also exhibited the obvious annual cycle of high in summer and low in winter. According to the criteria proposed by Duncan et al. (2010), $O_3$ formation regime in Shanghai was usually under $NO_x$-limited from June to August, and controlled by VOC-limited and transition regime for the rest of months. From May to September in 2014, Shanghai all under $NO_x$-limited regime, and $FNR_{SAT}$ reached the highest value over the past 10 years. In 2019, the value of

monthly $FNR_{SAT}$ fluctuated gently, which showed the trend moving towards the transition regime threshold of 1-2.

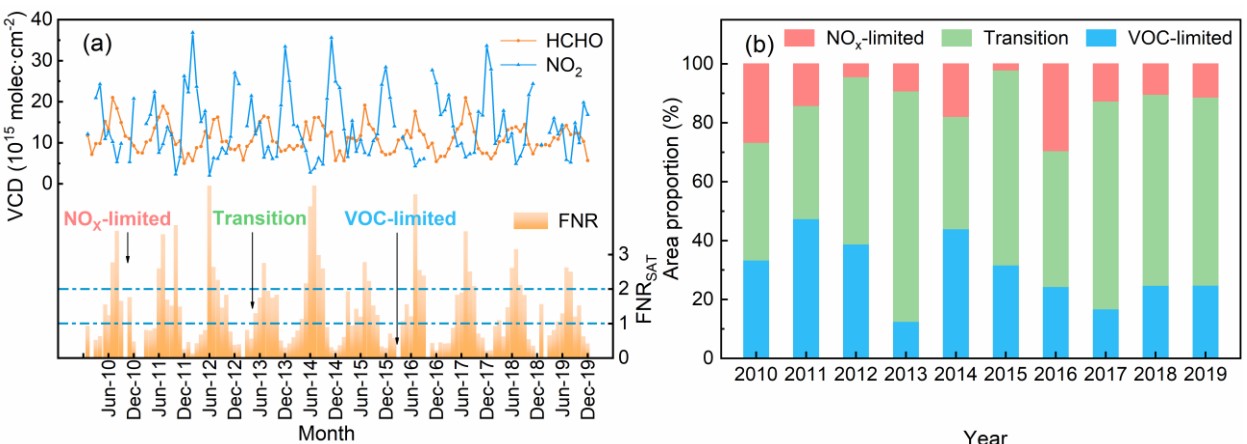

**Figure 5. Temporal and spatial variation of $FNR_{SAT}$ in Shanghai: (a) variations of HCHO, $NO_2$ and $FNR_{SAT}$ from 2010 to 2019, and (b) the area proportion for different $O_3$ formation regimes in these years.**

The spatial distribution of $FNR_{SAT}$ in Shanghai is presented in Fig. 6. Comparing $FNR_{SAT}$ in 2019 with that in 2010, the

$NO_x$-limited regime in the western Shanghai transformed into transition regime, and VOC-limited regime in the northern Chongming Island almost completely became transition regime. It manifests itself in the reduction in the $NO_x$-limited and VOC-limited regimes while the increase in the transition regime. Referring to previous study on the variation of $NO_2$ VCDs spatial distribution in Shanghai observed by satellite, this phenomenon may be related to the spatial characteristics of concentration variations of two precursors (Xue et al., 2020). This result is also reflected in Fig. 5b. In the past 10 years, the

proportion decreased from 33.4% to 24.9% for VOC-limited area, and from 26.7% to 11.3% for $NO_x$-limited area,

respectively. Meanwhile, the transition regime area increased from 40.0% to 63.8%. Xu et al. (2019) suggested that $O_3$ formation regime in Shanghai trend to transform from VOC-limited regime to $NO_x$-limited regime after 2020 through WRF-Chem model simulation. In this study, the increase of transition regime may be the transition state from VOC-limited to $NO_x$-limited regime.

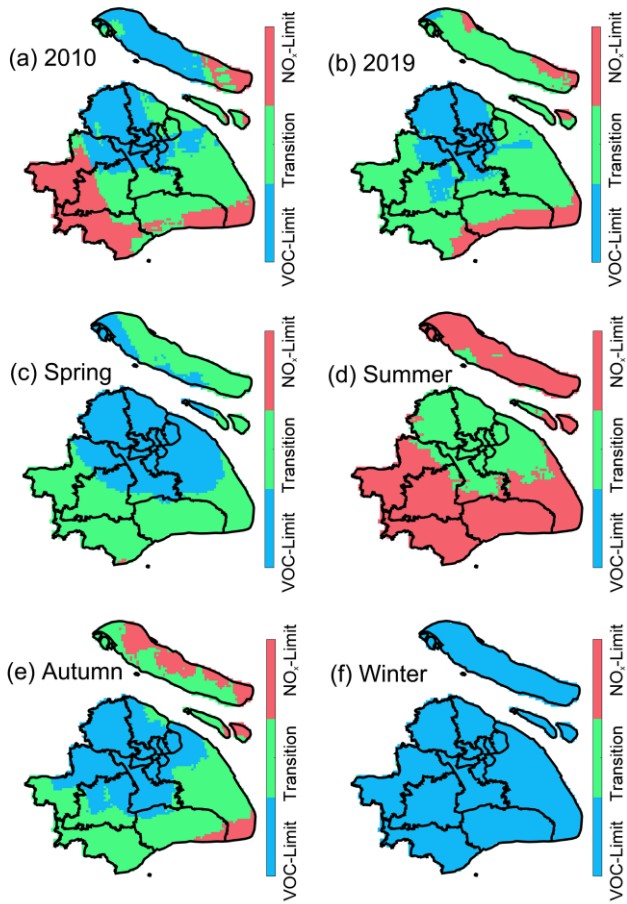


**Figure 6. The spatial distribution of FNR$_{SAT}$ for (a) 2010, (b) 2019, and (c) to (f) different seasons in Shanghai.**

Figure. 6c to Fig. 6f show the spatial distribution of FNR$_{SAT}$ for different seasons during the decade. In spring, the northern area of Shanghai was VOC-limited regime while the southern area was transition regime. From spring to summer, the VOC-limited area almost transformed into transition regime, and the transition regime nearly turned into $NO_x$-limited at the same

time, which mainly caused by the increase of HCHO and the decrease of $NO_2$. The distribution in autumn was similar to that in spring. In winter, Shanghai was basically under VOC-limited. In the light of the temporal and spatial distribution of $O_3$ formation regime inferred by FNR$_{SAT}$, the emission reduction measures for $O_3$ precursors would be more rationally.

Besides, there are differences in $O_3$ formation regime in urban and suburban areas of Shanghai, with the main manifestation that the central urban area was inclined to be in VOC-limited regime and the suburban area was more likely to be in $NO_x$-

limited regime. In order to analyse the differences in more detail, area within 10 km around the Jiangwan campus of Fudan University was selected to represent the urban area (about 12 km to the city center), and equally sized area around Dianshan Lake (31.09°N, 120.98°E) of Qingpu district was regarded as the suburban area (about 50 km to the city center), respectively. Since $O_3$ pollution is relatively serious from April to September, this period can be chosen as a research case. The maximum 8-hour average concentration of $O_3$ at the Hongkou and the Qingpu Dianshan Lake Site were used to characterize $O_3$

concentration in urban and suburban areas. $NO_2$ VCDs in urban area were higher than that in suburban, while HCHO VCDs presented the opposite character (Fig. 7a). The lower $NO_2$ in suburban area was associated with the less $NO_x$ emission than that in urban, corresponding to the larger $FNR_{SAT}$. From Fig. 7b, the $O_3$ production was under transition regime in urban area, while $NO_x$-limited regime occupied for most years in suburban area. The $O_3$ concentration in suburban area was always higher than that in urban (Fig. 7c). In this study, high concentration of $NO_x$ in urban area may lead to titration of $O_3$ by NO,

which would cause the lower $O_3$ concentration in urban (Geng et al., 2008; Duncan et al., 2010).

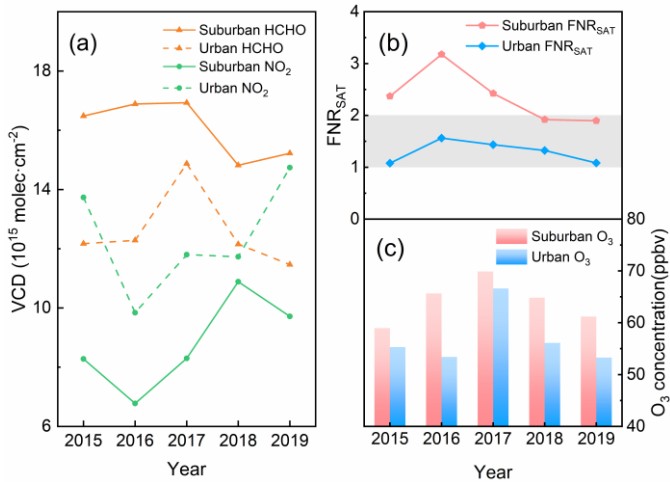

**Figure 7. Differences of (a) satellite observation of HCHO and $NO_2$ VCDs, (b) $FNR_{SAT}$ and (c) surface $O_3$ concentration in urban and suburban areas of Shanghai from 2015 to 2019.**

### 3.4. Correction of $FNR_{SAT}$

FNR is originally proposed as an indicator to characterize the sensitivity of the instantaneous $O_3$ production rate (Duncan et al., 2010). Satellite observation only reflects the averaged column of the trace gases around overpass time, so the $FNR_{SAT}$ may not accurately infer the surface $O_3$ formation regime during the day (Duncan et al., 2010; Jin and Holloway, 2015; Jin et al., 2017). $FNR_{SAT}$ was compared with $FNR_{LP}$ at the satellite overpass time on the monthly and daily scales. Results show that $FNR_{SAT}$ and $FNR_{LP}$ were consistent well on monthly scale ($R^2=0.95$, between April and August), but they were quite

different on the daily scale, and the relationship was easily affected by the boundary layer height (BLH) and other factors (Schroeder et al., 2017).

As shown in Fig. 8a to Fig.8c, LP-DOAS observation provided $FNR_{LP}$ with the high temporal resolution throughout the day, and $FNR_{LP}$ was in good agreement with hourly $O_3$ concentration. It means that $FNR_{LP}$ should be a good indicator to

distinguish the formation regime of surface $O_3$ on a detailed time scale. In addition, the highest value of $FNR_{LP}$ at noon means that $FNR_{SAT}$ at overpass time approximately close to the highest level in a day. Thus, it would be valuable to introduce the time series of $FNR_{LP}$ to make $FNR_{SAT}$ better reflect the characteristic of $O_3$ formation during the daytime.

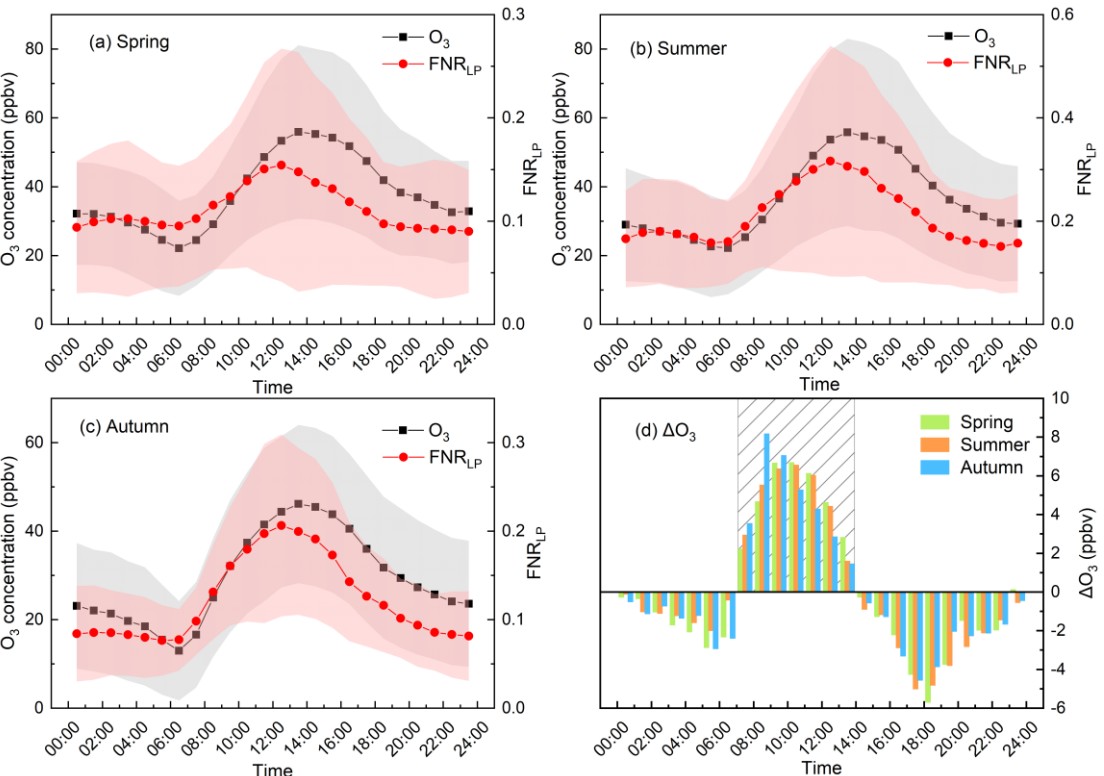

**Figure 8. Diurnal variations of $FNR_{LP}$, $O_3$ and $\Delta O_3$ for spring, summer and autumn during 2018-2019. $\Delta O_3$ represents increment of hourly $O_3$ concentration, as referred to Eq. (3). The shaded area in (d) represents the period when $\Delta O_3$ greater than 0.**

Three cases during the LP-DOAS observation at the Jiangwan campus of Fudan University were selected for the further discussion under the criteria of the hourly concentration of $O_3$ exceeding 200 μg/m³ (secondary concentration limit stipulated in ambient air quality standards of China, GB 3095-2012). In Fig. 9a and Fig. 9c, the upward trend in $FNR_{LP}$ as the $O_3$ concentration increases means that the formation of $O_3$ was under VOC-limited regime in Case 1 and 3. VOC-limited and transition regimes caught in Case 1 through $FNR_{SAT}$ were different from that identified by $FNR_{LP}$. In Case 3, the $O_3$ formation regime remained in VOC-limited regime, which was the same as $FNR_{LP}$. As shown in Fig. 8b, $FNR_{LP}$ indicated that the $O_3$ formation regime switched between three regimes in Case 2. As $O_3$ concentration increasing, the growth of $FNR_{LP}$ at beginning indicated VOC-limited regime, while the subsequent slow variation suggested transition regime. When $O_3$ concentration reached maximum, $FNR_{LP}$ got smaller, which referred to $NO_x$-limited regime. But $FNR_{SAT}$ only captured the VOC-limited and transition regime in Case 2. According to the results above, it is feasible and necessary to correct

FNR$_{SAT}$ to better represent the sensitivity of surface O$_3$ formation. Due to the less O$_3$ pollution in winter, FNR$_{SAT}$ was only corrected for the remaining three seasons.

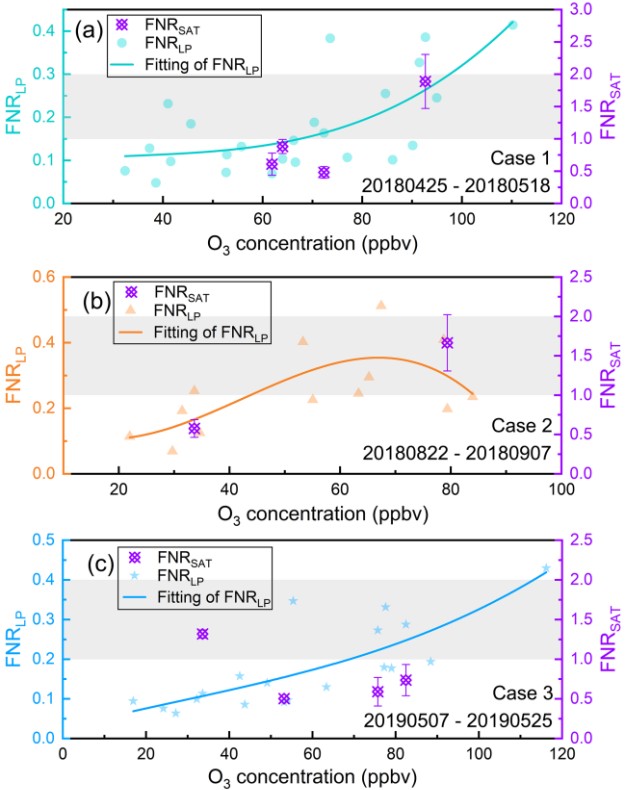

**Figure 9. Comparison of FNR$_{SAT}$ and FNR$_{LP}$ in three O$_3$ pollution cases. The curves are polynomial fit of FNR$_{LP}$, and the points are FNR$_{SAT}$ in corresponding cases, the gray area indicates the transition regime for FNR$_{SAT}$. Only LP-DOAS measured data between 08:00-18:00 LT were contained.**

The correction of FNR$_{SAT}$ was achieved through the following process. Firstly, the ratio of FNR at satellite overpass time to averaged FNR during O$_3$ pollution period ($\Delta$O$_3$ greater than 0, i.e. 07:00 to 13:00 LT, referred to Fig. 8d) obtained from ground surface measurement was assumed to be same with that of satellite observation, which is expressed as:

$$\frac{FNR_{SAT}}{\overline{FNR}_{SAT}} = \frac{FNR_{LP,overpass\ time}}{\overline{FNR}_{LP}} \qquad (1)$$

where $\overline{FNR}_{LP}$ is weighted average of FNR during O$_3$ pollution period observed by ground surface LP-DOAS measurement, FNR$_{LP,overpass\ time}$ is ground surface FNR observed at satellite overpass time. $\frac{FNR_{LP,overpass\ time}}{\overline{FNR}_{LP}}$ reflects the numerical relationship of FNR between satellite overpassed time and O$_3$ polluted period during a day, and can be serve as the correction coefficient to realize the correction for FNR$_{SAT}$ on time scale to obtain the $\overline{FNR}_{SAT}$. Considering the relationship

between time series of FNR and $O_3$ formation, $\Delta O_3$ was involved to calculate the weighted average of $FNR_{LP}$ ($\overline{FNR}_{LP}$) during 07:00 to 13:00 LT via Eq. (2), which can better indicate the ozone formation in a day.

$$\overline{FNR}_{LP} = \frac{\sum_{T=7}^{13} FNR_{LP,T} \times \Delta\Omega_{O3,T}}{\sum_{T=7}^{13} \Delta\Omega_{O3,T}} \qquad (2)$$

$$\Delta\Omega_{O3,T} = \Omega_{O3,T} - \Omega_{O3,T-1} \qquad (3)$$

Where $\Delta\Omega_{O3,T}$ is the increase of $O_3$ concentration at time T, $FNR_{LP,T}$ is the $FNR_{LP}$ at time T. Moreover, variation characteristics of $FNR_{LP}$ in different seasons (as shown in Fig. 8a to Fig. 8c) suggest that the correction of $FNR_{SAT}$ should be discussed seasonally.

Afterwards, the seasonal correction coefficients of 0.85, 0.84, and 0.77 were obtained for spring, summer, and autumn, respectively. It is noted that all the correct coefficients were less than 1 due to the $FNR_{LP}$ value for OMI overpasses time 310 relatively larger than other time. It would inevitably make the $O_3$ formation regime inferred by corrected $FNR_{SAT}$ trend to be VOC-limited. There were 87 months with effective $FNR_{SAT}$ in three seasons during 2010-2019 and the proportion of months for different regimes was listed in Table 1. Before the correction, the VOC-limited and $NO_x$-limited regimes, as well as transition regime were almost accounted for about one-third of the total months. After the correction, both of months for VOC-limited and transition regimes increased. The months under VOC-limited regime increased by 21.4% particularly, 315 while decreased about 25.9% for $NO_x$-limited regime. Jin et al. (2020) used $O_3$ exceedance probability as the indicator to analyse non-linear dependence of long-term surface $O_3$ concentration on precursor emissions, and determined the OMI $HCHO/NO_2$ under transition regime ranging from 3.2 to 4.1. These thresholds are significantly larger than the value proposed by Duncan et al. (2010), which has the equivalent effect as the use of correction factors less than 1 in this study.

Table 1. **Variations in the number and proportions of months in 2010-2019 for each regime before and after the correction.**

| Regime | Before corrected | Percentage before corrected | After corrected | Percentage after corrected | Percentage of change |
|---|---|---|---|---|---|
| VOC-limited | 28 | 32.2% | 34 | 39.1% | 21.4% |
| Transition | 32 | 36.8% | 33 | 37.9% | 3.1% |
| $NO_x$-limited | 27 | 31.0% | 20 | 23.0% | -25.9% |


In terms of spatial distribution, the $O_3$ formation regime after correction in different seasons is shown in Fig. 10. Compared with Fig. 6, part of the transition regime area in spring transformed into VOC-limited regime after the correction, VOC-limited regime obviously expanded and increased about 18.2%. Most area in Chongming Island transformed into VOC-limited regime. In summer, part of the $NO_x$-limited regime area transformed into transition regime, and the area of transition 325 regime increased about 16.2%. In autumn, the $NO_x$-limited regime almost disappeared, and the proportion of the transition

regime area also decreased significantly, for about 10.4%, while the proportion of the VOC-limited regime area increased about 21.5%. The O₃ formation regime in Chongming Island was basically under transition regime. Researchers also pointed that thresholds of criteria would be regionally dependent, so the local surface FNR and O₃ concentration should be introduced in satellite correction (Duncan et al., 2010; Jin et al., 2020).

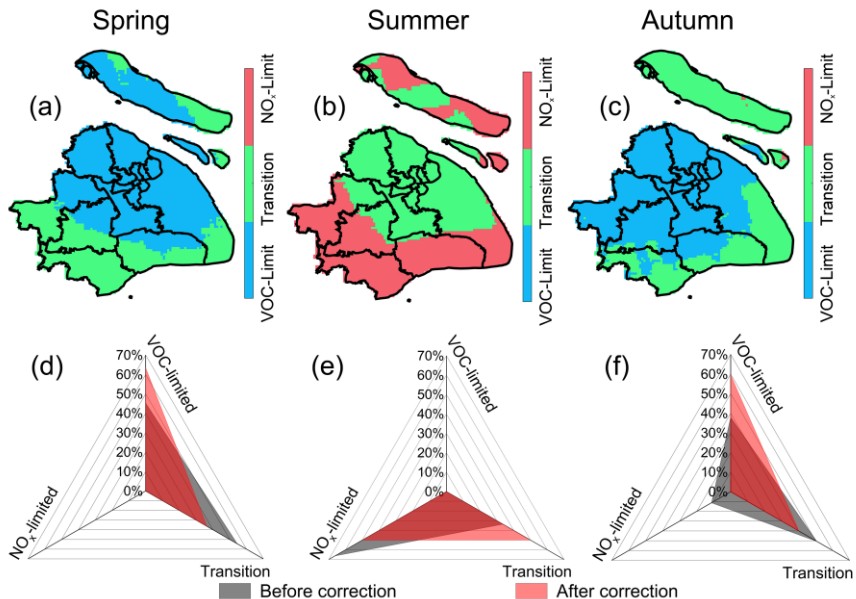


**Figure 10. Spatial distribution of O₃ formation regime (upper row) and area proportion for each regime (bottom row) after correction for different seasons in Shanghai during 2010-2019.**

In order to verify whether the correction of the satellite FNR improved the regime classification, O₃ formation regimes

determined by satellite FNR before and after the correction were compared with that of surface observation. The variations of O₃ with surface HCHO and NO₂ have been plotted to determine the O₃ formation regimes from the surface observation (Figure 11). The daytime surface HCHO and NO₂ are from LP-DOAS measurements, and the O₃ observed by SP-DOAS (short-path DOAS), which is also located at Jiangwan campus of Fudan University, have been used with a high temporal resolution. O₃ formation regimes inferred from satellite FNR before and after the correction have also been marked in Figure

11 separately.

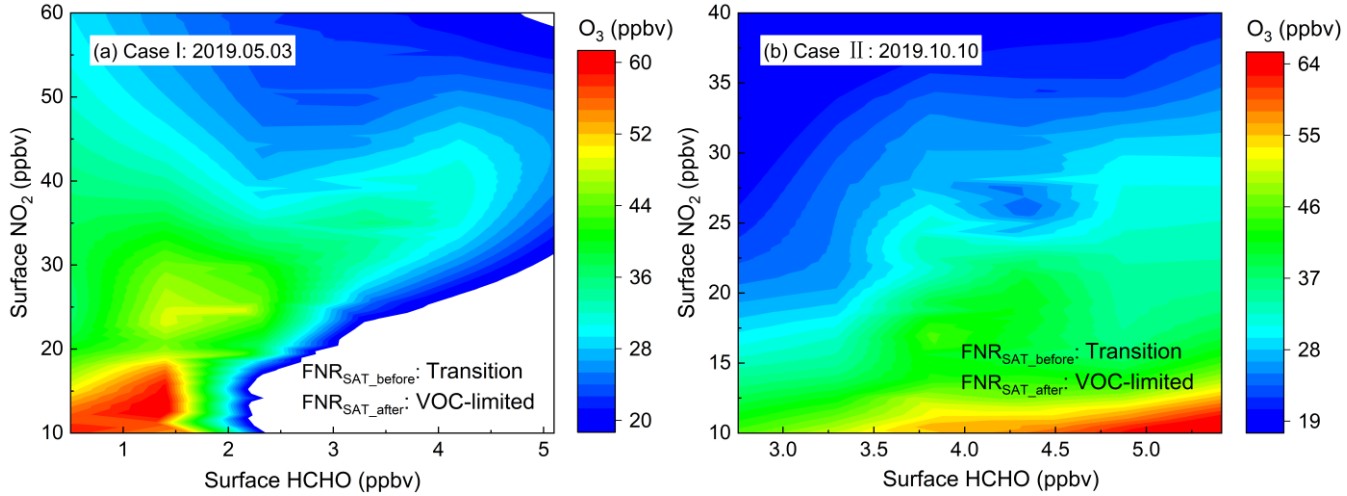

**Figure 11. The variation of O$_3$ with surface HCHO and NO$_2$ for two cases of (a) May 3th, 2019 and (b) October 10th, 2019. FNR$_{SAT\_before}$ and FNR$_{SAT\_after}$ indicate the O$_3$ formation regimes inferred from the satellite FNR before and after the correction.**


For Case I, O$_3$ decreases with the increase of NO$_2$, which can be attributed to the titration of O$_3$ by NO (Duncan et al., 2010). O$_3$ increased from top to bottom indicating it was under VOC-limited regime in Case I (Luo et al., 2020). For Case II, it can be seen that the high O$_3$ appeared with high HCHO and low NO$_2$, indicating it was under VOC-limited regime. The uncorrected satellite FNR indicated that these two cases were both under transition regime, while the corrected satellite FNR

indicated they transferred to VOC-limited regime, which are consistent with the results of surface observation. It indicates that the correction of FNR$_{SAT}$ in this study can be considered to be effective and make sense.

## 4. Summary and conclusions

Satellite data of OMI were used to study the temporal and spatial variation of HCHO in Shanghai from 2010 to 2019. HCHO VCDs fluctuated during the 10 years with obvious seasonal characteristics of highest value in summer, the lowest value in

winter, and the moderate level in spring and autumn. In terms of spatial distribution, HCHO VCDs in western area were much higher than that in eastern coastal area. Compared with 2010, HCHO VCDs of Shanghai in 2019 showed an overall downward trend. As for the influencing factors, temperature give the significant positive effect on HCHO VCDs while the abundant precipitation reduces HCHO in summer. Industry was an important contributor of HCHO, and anthropogenic secondary production of HCHO occupied the main part of the HCHO source.

In the past 10 years, O$_3$ formation regime changed toward transition regime gradually. O$_3$ formation regime in urban area was more likely to be VOC-limited regime, while regime in suburban area was more likely to be NO$_x$-limited. FNR$_{SAT}$ was corrected based on the hourly surface FNR and O$_3$ data to make it better to reflect O$_3$ formation in a day. After correcting FNR$_{SAT}$ with seasonal correction coefficients of 0.85, 0.84, and 0.77 for spring, summer, and autumn respectively, O$_3$

formation regime in Shanghai was more inclined to VOC-limited in both time and spatial distribution, and the effectiveness

of FNR$_{SAT}$ correction was confirmed by the surface observation. Thus, this correction is significant for using satellite data to improve the accuracy in indicating surface $O_3$ formation.

**Data availability.** Data are available for scientific purposes upon request to the corresponding author.

**Author Contributions:** DL and SW designed and implemented the research, and prepared the manuscript; RX contributed to the analysis of OMI products and MEIC inventory; JZ provided the HCHO and $NO_2$ data observed by LP-DOAS; SZ and ZS revised the manuscript; BZ provided constructive comments and supported the DOAS measurements.

**Competing interests.** The authors declare that they have no conflict of interest.


**Acknowledgments:** We acknowledge the free use of OMI HCHO and $NO_2$ products from NASA Goddard Earth Sciences (GES) Data and Information Services Center (DISC). We also thank Center for Earth System Science, Tsinghua University for MEIC data.

**Financial support:** This research was funded by National Key Research and Development Program of China (2017YFC0210002), National Natural Science Foundation of China (41775113, 21777026), and Funds for International Cooperation and Exchange of the National Natural Science Foundation of China (Grant No. 42061134006).

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
