# Peer review of "OMI-observed HCHO in Shanghai, China during 2010-2019 and ozone sensitivity inferred by improved HCHO/NO2 ratio"

_Atmospheric Chemistry and Physics, 2020_

## Author Comment (AC1)

**Response to reviewer #1**

We thank the reviewer for the constructive comments and suggestions, which are very positive to improve scientific content of the manuscript. We have revised the manuscript appropriately and addressed all the reviewer's comments point-by-point for consideration as below. The remarks from the reviewer are shown in black, and our responses are shown in blue color. All the page and line numbers mentioned following are refer to the revised manuscript without change tracked.

1. The novelty of this paper is the inclusion of ground-based observations, but I think the ground-based measurements are underused in this study. Since a main part of this study is on HCHO, I don't see how the authors use ground-based measurements of HCHO to support satellite HCHO. Do you see similar temporal patterns from ground vs. space? This may also help understand the difference between column vs. surface HCHO.

R: Thanks for the constructive comments. The ground surface HCHO has been measured by LP-DOAS at the Jiangwan campus of Fudan University in Shanghai (31.34 °N, 121.52 °E) during 2018-2019 to compare with satellite observation. In order to match ground-based and spaced observations for comparison, the satellite HCHO within 10 km of LP-DOAS measurement site were averaged as the satellite observation, while the surface HCHO observed by LP-DOAS between 13:00 and 14:00 were used considering the OMI overpasses time.

Figure R1 shows that HCHO VCDs and surface HCHO concentrations are not consistent very well. For daily observation, two observations are quite different, but the monthly averages are more similar. However, the difference still exists, for example, the highest value of HCHO VCD in 2019 appeared in June, while that of surface observations appeared in August. It also suggests that the FNR from satellite will deviate from the surface observed FNR.

It should be noticed that the vertical column density represents the concentration of the total column, while the LP-DOAS results only reflect the concentration near the ground. Previous studies show that HCHO is not completely concentrated near the ground, but has a high concentration at higher altitudes (Chan et al., 2019; Wang et al., 2019). It may explain the discrepancy of tropospheric HCHO VCD observed by satellite and the ground surface HCHO concentration. In addition, satellite data reflects the average level of a given area, while LP-DOAS is a single point measurement. The spatial heterogeneity of surface HCHO concentration in horizontal can also impact the consistency of this comparison.

We have also added the comparison between satellite and surface HCHO in the manuscript, please refer to Line 130-134.

[Figure]

*Figure R1. Comparison of daily and monthly HCHO observed by LP-DOAS and satellite. The black and red points represent the daily HCHO VCD and surface HCHO respectively, while the dot lines indicate the monthly averages.*

2.    There seems to be some artificial strip patterns with HCHO (Figure 2), which looks like due to the influence of OMI swath changes. It is not clear how the authors process OMI HCHO data. The authors mentioned they re-grid the data to 0.01°× 0.01°, which is much finer than the resolution of OMI. No details are provided in terms of spatial downscaling. In general, spatial oversampling is used to process OMI data to achieve better resolution (e.g. Zhu et al., 2014). I suggest the authors consider following such procedure.

R: Thanks for your professional comments. In this study, OMI HCHO data were processed through the following steps. Firstly, the targeted area was gridded into to a spatial resolution of 0.01°×0.01°, then the HCHO VCD of each pixel was assigned to the respective grid by determining the coordinates information. In addition, a weight function including cloud function and pixel size was introduced to effectively improve the quality of processed data (Xue et al., 2020). We have supplemented the data processing method in the revised manuscript, please refer to Line 89-91.

We have carefully reviewed the recommended article, as well as the articles that used the same strategy (Fioletov et al., 2011; McLinden et al., 2012; Zhu et al., 2014). In these studies, pixel falling within a certain radius of the grid center were averaged, and further was assigned to that grid. Then, the differences on HCHO VCDs between these two methods were compared. Three years, i.e. 2010, 2014 and 2018, were selected randomly for test, as shown in Figure R2. HCHO VCDs in Shanghai obtained by these two methods were almost the same in annual average. Most areas show similar HCHO levels, and about 84.6% ± 1.1% area of Shanghai shows the difference between the two

methods within 10%. Considering the different approaches in satellite data processing, such a level of difference is considered to be reasonable and accepted.

[Figure]

*Figure R2. Comparison of the spatial distribution of HCHO VCD in Shanghai for 2010, 2014 and 2018 obtained by these two processing methods. Method_1 represents the method recommended by the references, Method_2 represents the method used in this study.*

It can be noticed that there are strip patterns in the spatial distribution of HCHO VCD via Methond_2 in Figure R2 (also appeared in Figure 2), which not appear in the one via Methond_1. The absence of smoothing procedure in Method_2 may be the main reason causing the difference in the results of these two methods. After the improvements in the algorithm of OMI HCHO Version 3.0, across-track striping of the HCHO columns is a minor issue of the new satellite product (https://aura.gesdisc.eosdis.nasa.gov/data/Aura_OMI_Level2/OMHCHO.003/doc/README.OMHCHO.pdf). So the strip patterns in the spatial distribution of HCHO VCD are considered not introduced by artificial processing, but inherited due to unsmoothing in Method_2.

3. It's not clear to me how the authors explore the impacts of anthropogenic emissions on HCHO. There seems to be several issues. First, the authors only consider the primary emissions of HCHO, but a lot of HCHO is produced secondarily from other VOCs like alkene. The HCHO yield should also vary with VOC species, and also meteorology. Second, as I pointed out earlier, the authors did not consider the role of biogenic emissions especially isoprene. Without secondary HCHO, there is little we can learn about the driven factors of HCHO from this paper.

R: Thank you for the professional comments. Secondary production of HCHO from anthropogenic and biogenic sources has contributed greatly to HCHO, so it is necessary to consider the secondary production of HCHO (Zhu et al., 2017; Shen et al., 2019).

We have also considered the secondary production of HCHO in the revised manuscript, including anthropogenic and biogenic sources. Please refer to Line 176-205.

In order to get the secondary production of HCHO from anthropogenic sources, NMVOCs (Non-methane volatile organic compounds) emission inventory based on the SAPRC07 mechanism species from Multi-resolution Emission Inventory for China (MEIC) was used for years of 2010, 2012, 2014 and 2016. Secondary production of HCHO has been calculated based on 1-day HCHO yields of several NMVOCs under high-$NO_x$ condition (Shen et al., 2019). Table R1 summarizes the primary emissions and secondary productions of HCHO from different sectors of anthropogenic sources.

*Table R1. The primary emissions and estimated secondary productions of HCHO in Shanghai from anthropogenic NMVOCs based on SAPRC07 mechanism species.*

| Year | | Estimated HCHO production from each sector ($10^9$ g) | | | | |
|------|------|------|------|------|------|------|
| | | Industry | Power | Residential | Transportation | Total |
| 2010 | Primary[1] | 9.10 | 0.03 | 0.06 | 1.47 | 10.66 |
| | Secondary[2] | 240.58 | 0.52 | 15.14 | 66.04 | 322.28 |
| 2012 | Primary | 7.73 | 0.05 | 0.07 | 1.01 | 8.86 |
| | Secondary | 246.67 | 0.57 | 15.67 | 51.91 | 314.82 |
| 2014 | Primary | 6.88 | 0.05 | 0.07 | 0.74 | 7.74 |
| | Secondary | 253.32 | 0.50 | 16.44 | 44.32 | 314.58 |
| 2016 | Primary | 6.29 | 0.05 | 0.06 | 0.61 | 7.01 |
| | Secondary | 286.36 | 0.51 | 16.64 | 43.14 | 346.65 |

*[1] Primary indicates HCHO that is directly emitted by anthropogenic sources from MEIC inventory.*

*[2] Secondary indicates HCHO that is produced by anthropogenic NMVOCs, which is calculated based on 1-day HCHO yields.*

Regardless of the primary emissions or secondary productions of HCHO, industry sector corresponds to the largest yield, followed by transportation, residential, and the power. For the temporal pattern, the primary emission of HCHO keeps decreasing (about 34.2% compared to 2010), while secondary produced HCHO did not change significantly. The increase of secondary HCHO yields in 2016 was mainly due to the increased production from industry sector. In addition, the changes and proportional relationships between primary emission and secondary production of HCHO for different sectors are different, which suggests the VOCs source profiles of different sectors would affect the amount of secondary HCHO production.

In addition, considering the mechanism species for different chemical mechanisms in MEIC inventory may have impacts on HCHO secondary production, we have also tested the HCHO yields based on CB05 mechanism species. HCHO yields in eastern China during May-September 2010 were calculated based on CB05 and SAPRC07 separately, and results were compared with the study of Shen et al. (2019). As shown in Table R2, total HCHO yield based on CB05 is about 23% higher than that of SAPRC07, and the latter is much closer to the reference, with a deviation of 7.1%. The

high degree of lumping of PAR in CB05 may cause the corresponding individual VOC to be overestimated, and caused the large deviation of HCHO yield. Compared with CB05, SPRAC07 may be more accurate for calculating the HCHO yield. It also illustrates that the selection of chemical mechanism species would also introduce uncertainty to the estimation.

*Table R2. Estimated May-September total HCHO production in eastern China in 2010 from CB05 and SAPRC07 mechanism species and the comparison with reference.*

| Species | Estimated HCHO production from anthropogenic NMVOCs in eastern China (Tg) | | |
|---------|-------|---------|-----------|
| | CB05[1] | SAPRC07 | Reference[2] |
| Ethane | 0.27 | 0.28 | 0.28 |
| Propane | 1.29 | 0.19 | 0.54 |
| ≥C4 alkanes | 1.40 | 1.75 | 1.87 |
| Ethylene | 0.76 | 0.75 | 0.75 |
| ≥C3 alkenes | 0.89 | 1.88 | 1.11 |
| Benzene | 0.03 | 0.05 | 0.038 |
| Toluene | 0.51 | 0.35 | 0.37 |
| Xylenes | 0.62 | 0.23 | 0.11 |
| Formaldehyde | 0.14 | 0.09 | 0.09 |
| Acetaldehyde | 0.11 | 0.04 | 0.04 |
| Methanol | 0.06 | 0.06 | 0.06 |
| Ethanol | 0.06 | - | 0.01 |
| Acetone | 0.90 | 0.15 | 0.16 |
| Methy ethyl ketone | 0.14 | 0.04 | 0.04 |
| Isoprene | 0.01 | 0.01 | 0 |
| Monoterpene | 0.02 | 0.02 | 0 |
| Total | 7.21 | 5.86 | 5.47 |

*[1] Highly lumping mechanism species of CB05, including PAR, are approximately allocated through some individual VOC concentrations observed locally.*
*[2] Reference refer to the estimated May-September total HCHO production from Anthropogenic emission in eastern China averaged during 2005-2016 (Shen et al., 2019).*

HCHO yield from biogenic sources can be estimated from BVOCs emission inventory. Model of Emissions of Gases and Aerosols from Nature (MEGAN) is widely used to simulate the emission of BVOCs. As we currently cannot use MEGAN to accurately simulate four-year (2010, 2012, 2014, 2016) BVOCs emissions, we have used the annual total BVOCs emissions of Shanghai in 2014 (about $1.2 \times 10^4$ t) for the estimation (Liu et al., 2018a; Liu et al., 2018b). Isoprene, methanol and monoterpenes were dominant compositions of BVOCs and accounted about 81.3% of the total. We have calculated HCHO yields contributed by isoprene, methanol and monoterpenes, as shown in Table R3.

*Table R3. The annual BVOCs emissions and HCHO yields over Shanghai in 2014.*

| BVOC | Emission ($10^9$ g) | HCHO yield ($10^9$ g) |
|---|---|---|
| Isoprene | 4.63 | 4.70 |
| Methanol | 4.26 | 3.99 |
| Monoterpenes | 0.86 | 0.38 |
| Total | 9.75 | 9.07 |

Accordingly, HCHO yield from BVOCs emission was estimated to be about $9.07 \times 10^9$ g, and mostly produced from isoprene and methanol. The calculated HCHO yield from BVOCs emission is similar to that of previous study during 2005-2016 (Shen et al., 2019). In addition, compared with anthropogenic sources, HCHO yield from BVOCs is much smaller, which indicates that the anthropogenic is the main contributor of the secondary production of HCHO in Shanghai (Shen et al., 2019; Fan et al., 2021).

As shown in Table R4, we have also reviewed the related studies about the BVOCs emissions in Shanghai and its surrounding areas (the Yangtze River Delta) in relevant years. Wang et al. (2021a) assessed the impacts of land cover change and climate variability on BVOCs emissions in China from 2001 to 2016, in which variations of BVOCs emissions in Shanghai over the years were extremely small. Considering the different input dataset and settings would bring large differences in the simulated results, it was unfeasible to use BVOCs emissions from different studies for the investigation of temporal variation. Therefore, the BVOCs emissions in 2014 were used to basically characterize the approximate level of BVOCs from 2010 to 2016 in this study.

*Table R4. Comparison of simulated BVOCs emissions in Shanghai (SH) and the Yangtze River Delta (YRD) based on MEGAN.*

| Simulated year | Reference | MEGAN version | Region | BVOCs emission ($10^4$ t) |
|---|---|---|---|---|
| 2010 | Song et al. (2012)[1] | V 2.04 | YRD | 110 |
| | | | SH | 0.122 |
| 2014 | Liu et al. (2018a; 2018b)[2] | V 2.10 | YRD | 188.6 |
| | | | SH | 1.2 |
| 2016 | Wang et al. (2021b)[3] | V 3.1 | YRD | 162 [1] |
| | | | SH | ~ 0.34 [4] |

*[1] Total annual emission inferred from the simulated BVOCs emissions in January, April, July and October.*
*[2] A variety of methods were used to reduce the uncertainty of plant functional types (PFT) database. The proportions of dominant components of BVOCs were also provided.*
*[3] BVOCs emission was simulated without drought stress.*
*[4] It is BVOCs emissions in July, which has been inferred from Fig. S3 of Wang et al. (2021b).*

As mentioned in Reviewer #2, HCHO yield was also impacted by the $NO_x$ levels, e.g. $RO_2$ radical from VOCs react with $HO_2$ to from organic peroxides under low $NO_x$

condition. This process reduces the reaction of $RO_2$ and NO, which in turn decreases the production of HCHO, therefore, HCHO yield from VOCs is proportional to $NO_x$ condition (Palmer et al., 2006; Marais et al., 2012; Miller et al., 2017). In this study, the estimation using a fixed HCHO yield may overestimate HCHO production in later years due to the decreases of $NO_x$ in Shanghai (Xue et al., 2020). In previous studies, the proportional relationship between HCHO yield and $NO_x$ condition was usually obtained when 1 ppbv of $NO_x$ regard as the high condition, and 0.1 ppbv of $NO_x$ regard as low condition (Palmer et al., 2006; Marais et al., 2012; Miller et al., 2017). However, the $NO_x$ concentration in Shanghai is still relatively higher (basically 30-60 ppbv in urban) (Gao et al., 2017). Therefore, in such a high $NO_x$ condition, the effect of $NO_x$ decreases on HCHO yield needs to be further studied.

4. Recent literature report there is uncertainty with the regime threshold for HCHO/NO2. The authors consider the uncertainty with diurnal cycle, but even at the overpass time, the regime threshold may also vary (Shroeder et al., 2017; Jin et al., 2020; Souri et al., 2020). I suggest the authors be more cautionary about applying the thresholds to separate regimes. More validation analysis is needed to support their regime classification.

R: Thank you for the professional comments. We have carefully reviewed the recommended articles. In these articles, various regime thresholds were proposed through chemical model or combining satellite result with ground surface observation, the complexity of applying the threshold is also discussed in detail (Schroeder et al., 2017; Jin et al., 2020; Souri et al., 2020). Although the $O_3$ formation sensitivity cannot be perfectly inferred from satellite HCHO/$NO_2$, this method benefits from the advantages of satellite observation and can still provide useful information on $O_3$ formation.

In this study, we did not obtain the new regime thresholds through such method, but used LP-DOAS data to correct the value of FNR observed by satellite. These two programs are similar in actual effect. Please refer to the following response for a detailed explanation of satellite FNR correction.

5. As I commented previously, the correction for diurnal variation doesn't make sense to me. First, the authors did not consider the difference between column-based satellite HCHO/NO2 vs. surface observed HCHO/NO2. Given the variation of the boundary layer height, the relationship between surface and column HCHO/NO2 should also vary with time. Second, it's not clear to me why the authors use ΔO3 to weight FNR. If the authors are only interested in the time when ozone production is most efficient, wouldn't it be easier to look the 1-hour maximum ozone? Third, there is no evidence supports whether such changes actually improved the regime classification.

R: Thank you for the professional comments. At the beginning, we have verified the relationship between surface FNR and $O_3$ during the day. As shown in Figure R3, the hourly $O_3$ concentration is in good agreement with surface FNR, which means that the surface FNR observed by LP-DOAS can be a good indicator of ground surface $O_3$ on a detailed time scale. Considering that satellites can only characterize the FNR situation

at overpass time, while LP-DOAS can provide a longer period of surface FNR variations, we hope to introduce the time series of LP-DOAS FNR to make satellite FNR be a better indicator that can reflect the characteristic of $O_3$ formation during the daytime.

[Figure]

***Figure R3. Diurnal variations of surface FNR observed by LP-DOAS and $O_3$ for different seasons during 2018-2019.***

Then, we have compared the satellite FNR and surface FNR observed by LP-DOAS on daily and monthly scales, and discussed whether satellite FNR can indicate ground surface $O_3$ properly like LP-DOAS FNR. In the following comparison, surface FNR is observed by LP-DOAS for 13:00-14:00, satellite FNR is the average value of 10 km area around LP-DOAS measurement site.

Figure R4 shows the variation of monthly satellite and surface FNR. Since the satellite and surface FNR represent the column average and ground surface $HCHO/NO_2$ respectively, they have a significant difference in the numerical value, the satellite FNR is significantly larger than the surface FNR. However, a strong correlation between the monthly satellite and surface FNR ($R^2 = 0.95$) was found from April to August, which means that satellite FNR can characterize the $O_3$ formation on monthly scale like surface FNR.

[Figure]

*Figure R4. The variations of monthly satellite and surface FNR at the location of LP-DOAS for April to August during 2018-2019.*

For the daily comparison, we have followed the suggestion to consider the influence of the boundary layer height (BLH) on the relationship between the satellite and surface FNR. The BLHs at 13:00 local time was selected to conform the overpass time of satellite. Figure R5 plotted the satellite and surface FNR under different BLHs. It indicates that on the daily scale, satellite and the surface FNR were quite different, and the relationship is affected by BLH. When the atmospheric turbulence is strong, the boundary layer is higher than 1500 m, the fitting slope of the satellite and surface FNR is small (slope=0.59, R=0.80). However, in the case of weak atmospheric turbulence, the corresponding boundary layer is lower than 1000 m, and the fitting slope of the satellite and surface FNR is large (slope=2.59, R=0.67) in this situation.

[Figure]

*Figure R5. Satellite and surface FNR under different boundary layer heights. The gray dashed line represents y=x, the red and blue lines represent the fit of satellite and surface FNR when the BLHs greater than 1500 m and less than 1000 m respectively. The BLH data comes from the fifth generation European Centre for Medium-Range Weather Forecasts reanalysis dataset (https://cds.climate.copernicus.eu/cdsapp#!/home).*

Regarding the parameter to weight FNR, we are not only interested in the time when $O_3$ formation is most efficient. 1-hour maximum $O_3$ only represents $O_3$ condition in a moment, which is contrary to the aim to make the corrected FNR better reflecting the temporal formation of $O_3$ during the daytime by introducing the time series of LP-DOAS FNR. In addition, as FNR is originally proposed as an indicator to characterize the instantaneous $O_3$ production rate, we used $\Delta O_3$ as the weight to avoid the effect of $O_3$ accumulation (high $O_3$ concentration but with small increase or even decrease of $O_3$) (Duncan et al., 2010). Take June 5th, 2018 as an example, the 1-hour maximum $O_3$ in Hongkou Site was about 109.35 ppbv, while $\Delta O_3$ was only 0.47 ppbv at that time, indicating the low efficient of $O_3$ formation. While from 07:00 to 12:00 that day, $\Delta O_3$ were basically greater than 10 ppbv. Therefore, we used the hourly $\Delta O_3$ as the weight to correct satellite FNR combined with the hourly LP-DOAS FNR.

In order to verify whether the correction of the satellite FNR improved the regime classification, we have compared the $O_3$ formation regimes determined by satellite FNR before and after the correction with that of surface observation. The variations of $O_3$ with surface HCHO and $NO_2$ during the daytime have been plotted to determine the $O_3$ formation regimes from the surface observation (Figure R6). The surface HCHO and $NO_2$ are from LP-DOAS measurements, and the $O_3$ observed by SP-DOAS (short-path differential optical absorption spectroscopy), which also located at Jiangwan campus of Fudan University, have been used with a finer temporal resolution. $O_3$ formation regimes inferred from satellite FNR before and after the correction have also been marked in Figure R6 separately.

[Figure]

*Figure R6. The variation of $O_3$ with surface HCHO and $NO_2$ for two cases of (a) May 3th, 2019 and (b) October 10th, 2019. $FNR_{SAT\_before}$ and $FNR_{SAT\_after}$ indicate the $O_3$ formation regimes inferred from the satellite FNR before and after the correction.*

For Case R1, $O_3$ decreases with the increase of $NO_2$, which can be attributed to the titration of $O_3$ by NO (Duncan et al., 2010). $O_3$ increased from top to bottom of the diagram indicating it was under VOC-limited regime in Case R1 (Luo et al., 2020). For Case R2, it can be seen that the high $O_3$ appeared with high HCHO and low $NO_2$, indicating it was under VOC-limited regime. The uncorrected satellite FNR indicated that these two cases were both under transition regime, while the corrected satellite FNR indicated they transferred to VOC-limited regime, which are consistent with the

results of surface observation. Therefore, the correction of satellite FNR can be considered to be effective and make sense.

All these discussed above has been method in the revised manuscript, please refer to the Line 258-348.

Minor Comments:

1.  Lines 116 to 120: Do you see similar seasonal cycle of HCHO from ground?

R: Thanks for the comment. As shown in Figure R1, the monthly averages of ground surface HCHO at 13:00-14:00 observed by LP-DOAS also show the similar seasonal cycle with satellite observation, that high in summer and low in winter. In addition, we have also calculated the averaged ground surface HCHO for the whole day, result also shows similar seasonal cycle (Figure R3). The seasonal ground surface HCHO concentrations is $2.25 \pm 0.40$ ppbv in spring, $3.08 \pm 0.51$ ppbv in summer, $2.47 \pm 0.61$ ppbv in autumn and $2.21 \pm 0.88$ ppbv in winter, respectively. We have added it in the revised manuscript, please refer to the Line 133-134. The division of seasons is referred to the response below. We have also added Figure R3 into Figure 1 of the revised manuscript to characterize the seasonal variation of ground surface HCHO.

[Figure]

*Figure R3. Seasonal averaged ground surface HCHO concentrations observed by LP-DOAS during 2018-2019.*

2.  Figure 1: Please define season here.

R: Thanks for the suggestion. In the study, seasons were divided as March, April, and May for spring, June, July, and August for summer, September, October, and November for autumn, December, January and February for winter. We have also defined seasons in the revised manuscript, please also refer to Line 120-122.

3.   Figure 1: I'd suggest include error bars to indicate spatial variation.
R: Thanks for the suggestion. We have added error bars to Figure 1 to indicate the spatial variation. The seasonal variation of ground surface HCHO was also added as Figure 1(d). Please refer to the updated Figure 1 in the revised manuscript.

[Figure]

*Figure R4. OMI and LP-DOAS observed time series of HCHO in Shanghai. (a) to (c) reflect the annual, monthly and seasonal variations of HCHO VCD during 2010-2019, error bars indicate the spatial variation of HCHO VCD. (d) reflects the monthly variation of surface HCHO observed by LP-DOAS during 2018-2019. (Figure 1 in the manuscript).*

4.   Figure 3: Why did you choose to show seasonal cycle only? I think it will be more interesting if you can show the time series from 2010 to 2018, and see how HCHO is correlated with each factor. This may also help explain the inter-annual variability of HCHO.
R: Thank you for the comments. We have followed the suggestion and analyzed the relationship between time series of HCHO VCDs and meteorological variables including temperature, sunshine hours, precipitation, and relative humidity via the linear regression. The stepwise regression results show that, the correlation between meteorological variables and HCHO VCD is not significant except temperature. Therefore, only the linear regression of temperature contribution and HCHO VCD was shown in Figure R4. We have presented Figure R4 in the manuscript as Figure 3, and moved the original Figure 3 to supplement and marked it as Figure S2.

[Figure]

*Figure R4. Monthly HCHO VCDs and the temperature contribution in Shanghai during 2010-2018. (a) reflects the temporal variations and (b) illustrates the correlation analysis. The black points represent the annual average residuals.*

The temperature contribution was strongly correlated with the observed HCHO VCD ($R^2 = 0.72$), which means that temperature can explain about 72% of the variation of HCHO VCD. The remaining part that cannot be explained by temperature appears in the form of residual, which is considered as the influences of other changing factors such as anthropogenic emissions (Li et al., 2019). We have also noticed that the residual in summer in some years would be particularly large, which indicates that in addition to temperature, there are other factors affecting HCHO VCD significantly in summer. Therefore, we have further analyzed the phenomenon in June that the HCHO VCD declines when the temperature rises, the precipitation and relative humidity rises significantly, as shown in Figure S2. Shanghai has a subtropical monsoon climate with rain and heat in the same period. Precipitation and relative humidity surged in June, while HCHO VCD decreased slightly with the increase of temperature. High relative humidity in July favoured the wet deposition of HCHO largely and offset the impact of rising temperature, resulting in a small decrease in HCHO VCDs.

We have regarded the changes of monthly averaged relative humidity and HCHO VCD as two variables, as shown in Eq. (R1) (R2).

$$\Delta RH_i = RH_i - RH_{i-1} \qquad (R1)$$
$$\Delta VCD_i = VCD_i - VCD_{i-1} \qquad (R2)$$

where $\Delta RH_i$ and $\Delta VCD_i$ are the changes of relative humidity and HCHO VCD for month i (Jun, Jul and Aug) between 2010 and 2018, respectively.

Then, the response of monthly HCHO VCDs variations to the changes of monthly relative humidity has been examined by Fisher's exact test (Clinton et al., 2020). We have defined the $\Delta VCD_i$ exceeding $\pm 1 \times 10^{15}$ molec·cm$^{-2}$ (about 10% of VCD$_{range}$) as an obvious increase or decrease of HCHO VCDs. And two conditions of 10% and 20% changes on RH$_{range}$ were checked. The RH$_{range}$ and VCD$_{range}$ is the range of relative humidity and HCHO VCDs between respective maximum and minimum, which are represented by the subscripts of 'max' and 'min' respectively, as expressed in Eq. (R3) and (R4).

$$RH_{range} = RH_{max} - RH_{min} \qquad (R3)$$
$$VCD_{range} = VCD_{max} - VCD_{min} \qquad (R4)$$

Results of Fisher's exact test determined that when $\Delta RH_i$ exceeds 10% of RH$_{range}$, no

significant correlation is found between these two variables (P = 0.637); when the change degree goes up to 20%, there is a significant correlation between them (P < 0.05). Therefore, it can be inferred that the variation in HCHO VCDs is related to the significant change of relative humidity in summer.

All these discussed above has been method in the revised manuscript, please refer to the Line 156-175.

5.  Figure 4: Need to include secondary HCHO from both anthropogenic and biogenic VOCs.

R: As replied above, the secondary HCHO from anthropogenic and biogenic VOCs have been estimated. Considering the lack of continuous BVOCs emission data, we only have plotted the primary emission and secondary production from anthropogenic sources. The results are shown in Figure R5 (Figure 4 in the revised manuscript). The biogenic contribution of HCHO and its uncertainty were discussed in the manuscript, and result in 2014 was marked as red cross in Figure R5(b). Please refer to Line 182 for detailed.

[Figure]

*Figure R5. Primary emissions and estimated secondary productions of HCHO from anthropogenic NMVOCs, biogenic contribution of HCHO in 2014 was marked as red cross in Figure R5(b).*

6.  Figure 7: How do you define urban vs. rural areas?

R: Thanks for the comment. We have defined the urban and rural areas according to its distance from the city center of Shanghai (31.24 °N, 121.48 °E). The distances from Jiangwan campus of Fudan University and Dianshan Lake to the city center are about 12 km and 50 km, respectively. Considering the civilization and development of Shanghai whole city, it would be more accurate to use suburban area instead of rural area. Therefore, we have taken areas around Jiangwan campus of Fudan University, which is closer to the city center, as the representative of the urban, and areas around Dianshan Lake, which is much far from the city center, as the representative of the suburban. The distances away from city center were also supplemented in the manuscript. Please refer to Line 242-244.

7.  Figure 8: It's unclear whether you're showing FNR and ozone for one site or three sites together? If one site, which site?

R: Thanks for the comments. Figure 8 showed the three cases only observed at the

Jiangwan campus of Fudan University. We have clarified it and please refer to Line 274-276.

---

## Author Comment (AC2)

**Response to review #2**

We thank the reviewer for the constructive comments and suggestions, which are very positive to improve scientific content of the manuscript. We have revised the manuscript appropriately and addressed all the reviewer's comments point-by-point for consideration as below. The remarks from the reviewer are shown in black, and our responses are shown in blue color. All the page and line numbers mentioned following are refer to the revised manuscript without change tracked.

This study examined long-term HCHO columns in Shanghai with OMI and ground-based remote sensing observations. The author also studied the ozone sensitivity in Shanghai and proposed a correction factor to satellite HCHO to NO2 ratio. The authors found that applying such a correction results in ozone formation is more likely to be under the VOC-limited regime. In my view, this work is among few studies exclusively focusing on HCHO observations and ozone sensitivity in China, thus is appropriate for publication at ACP subject to the following concerns.

1. The authors examined monthly averaged columns at a spatial resolution of 0.01°×0.01° degree. I am not sure whether there would be enough pixels in each grid cell to reduce the uncertainty in mean HCHO column to a much lower and acceptable level. This could be a potential issue in the winter seasons when SZA is high and the light path is long.

R: Thanks for your professional comments. In order to check whether there were enough pixels in each grid, we have counted the number of days that each grid has been assigned with HCHO VCDs between 2010 and 2019 on monthly scale. Here, we have defined the effective observation days (EODs) as the number of days that once any grid within Shanghai areas has been still designated after the data quality filtering. Afterwards, we have calculated the percentages of areas with observed days reached 60% EODs for every month during 2010-2019, results are shown in Table R1. For example, in January, about 80% areas of Shanghai have been assigned with HCHO VCDs during more than 60% of the total EODs.

*Table R1. Proportion of areas with observed days reached 60% EODs in each month of 2010-2019.*

| Month | Proportion | Month | Proportion |
|---|---|---|---|
| January | 79.9% ± 24.6% | July | 72.2% ± 31.8% |
| February | 61.6% ± 33.6% | August | 69.7% ± 25.7% |
| March | 87.7% ± 16.5% | September | 71.2% ± 33.4% |
| April | 94.3% ± 9.1% | October | 59.7% ± 30.9% |
| May | 86.1% ± 30.2% | November | 88.5% ± 13.3% |
| June | 33.4% ± 25.1% | December | 84.4% ± 18.6% |

Table R1 reveals that the winter months did not show the obvious low proportion, while

the proportion in June is much lower. It can be explained by the impacts of abundant precipitation in June (as shown in Figure S2), which leads to only a small amount of areas were assigned with HCHO VCD on the EOD after screening by cloud fraction. So, we considered that the monthly averaged HCHO VCDs can be the representative of the mean level of HCHO in Shanghai when the temporal patterns were discussed in the manuscript. While the spatial characteristics have been investigated with high spatial resolution only when more EODs have been oversampled, e.g. in seasonal with multiple years averaged.

2.  Did the authors correct the well-known drift in OMI SAO HCHO product? If not, please do so and update the results accordingly.

R: Thanks for your suggestions. Previous studies mentioned that OMI-SAO HCHO columns display significant drift due to instrument aging (Marais et al., 2012; Zhu et al., 2014; Zhu et al., 2017), in which OMI HCHO Version 2.0 data were used. However, the updated product (OMI HCHO Version 3.0) were used in this study, which has a significant improvement in treating the increasing trend of background values (https://aura.gesdisc.eosdis.nasa.gov/data/Aura_OMI_Level2/OMHCHO.003/doc/READ ME.OMHCHO.pdf).

In order to check whether there is still obvious drift, we have performed temporal linear regression of the deseasonalized monthly averaged HCHO columns during 2010-2019 for remote pacific region (29°-33°N, 160°-140°W) around the same latitude as Shanghai (Zhu et al., 2017). Zonal mean HCHO columns have been calculated with 0.5° latitude steps. It was found that there was no obvious linear trend for the deseasonalized HCHO VCDs on monthly scale in those eight zonal latitude ranges ($R^2$ ranged from 0.06 to 0.19 with average of 0.11), suggesting that the updated OMI HCHO product used in this study does not include the obvious drift. Figure R2 shows the time series of deseasonalized zonal mean HCHO VCDs for the highest and lowest latitudes on monthly scale. We have clarified in the manuscript, please refer to Line 99-101.

[Figure]

*Figure R2. The time series of deseasonalized zonal monthly mean HCHO VCDs of different latitudes in the remote pacific region (29°-33°N, 160°-140°W). Monthly variations of zonal mean*

*HCHO VCD with the maximum and minimum latitude are exhibited as the example (29.25°N and 32.75°N stands for range of 29.0°-29.5°N and 32.5°-33.0°N, respectively).*

3. In page 7, please clarify how to get the HCHO emission rate? Is HCHO mainly secondary? If so, how to determine HCHO yields from NMVOCs? The authors may also want to consider the lower HCHO yields from NMVOCs as NOx emissions drop in the study period.

R: Thank you for the professional comments. Figure 7 in the original manuscript reflects the annual anthropogenic HCHO primary emissions and does not include the secondary production of HCHO. However, secondary production of HCHO from anthropogenic and biogenic sources has contributed greatly to HCHO (Zhu et al., 2017; Shen et al., 2019). As suggested by Reviewer #1, we have also considered the secondary production of HCHO in the revised manuscript, including anthropogenic and biogenic sources. Please refer to Line 176-205.

In order to get the secondary production of HCHO from anthropogenic sources, Non-methane volatile organic compounds (NMVOCs) emission inventory based on the SAPRC07 mechanism species from Multi-resolution Emission Inventory for China (MEIC) was used for years of 2010, 2012, 2014 and 2016. Secondary production of HCHO has been calculated based on 1-day HCHO yields of several NMVOCs under high-$NO_x$ condition (Shen et al., 2019). Table R2 summarizes the primary emissions and secondary productions of HCHO from different sectors of anthropogenic sources.

*Table R2. The primary emissions and estimated secondary productions of HCHO in Shanghai from anthropogenic NMVOCs based on SAPRC07 mechanism species.*

| Year | | Estimated HCHO production from each sector ($10^9$ g) | | | | |
|---|---|---|---|---|---|---|
| | | Industry | Power | Residential | Transportation | Total |
| 2010 | Primary[1] | 9.10 | 0.03 | 0.06 | 1.47 | 10.66 |
| | Secondary[2] | 240.58 | 0.52 | 15.14 | 66.04 | 322.28 |
| 2012 | Primary | 7.73 | 0.05 | 0.07 | 1.01 | 8.86 |
| | Secondary | 246.67 | 0.57 | 15.67 | 51.91 | 314.82 |
| 2014 | Primary | 6.88 | 0.05 | 0.07 | 0.74 | 7.74 |
| | Secondary | 253.32 | 0.50 | 16.44 | 44.32 | 314.58 |
| 2016 | Primary | 6.29 | 0.05 | 0.06 | 0.61 | 7.01 |
| | Secondary | 286.36 | 0.51 | 16.64 | 43.14 | 346.65 |

*[1] Primary indicates HCHO that is directly emitted by anthropogenic sources from MEIC inventory.*

*[2] Secondary indicates HCHO that is produced by anthropogenic NMVOCs, which is calculated based on 1-day HCHO yields.*

Regardless of the primary emissions or secondary productions of HCHO, industry sector corresponds to the largest yield, followed by transportation, residential, and the power. For the temporal trend, the primary emission of HCHO keeps decreasing (about 34.2% compared to 2010), while secondary produced HCHO did not change

significantly. The increase of secondary HCHO yields in 2016 was mainly due to the increased production from industry sector. In addition, the changes and proportional relationships between primary emission and secondary production of HCHO for different sectors are different, which suggests the VOCs source profiles of different sectors would affect the amount of secondary HCHO production.

HCHO yield from biogenic sources can be estimated from BVOCs emission inventory. Model of Emissions of Gases and Aerosols from Nature (MEGAN) is widely used to simulate the emission of BVOCs. As we currently cannot use MEGAN to accurately simulate four-year (2010, 2012, 2014, 2016) BVOCs emissions, we have used the annual total BVOCs emissions of Shanghai in 2014 (about $1.2 \times 10^4$ t) for the estimation (Liu et al., 2018a; Liu et al., 2018b). Isoprene, methanol and monoterpenes were dominant compositions of BVOCs and accounted about 81.3% of the total. We have calculated HCHO yields contributed by isoprene, methanol and monoterpenes, as shown in Table R3.

*Table R3. The annual BVOCs emissions and HCHO yields over Shanghai in 2014.*

| BVOC | Emission ($10^9$ g) | HCHO yield ($10^9$ g) |
|---|---|---|
| Isoprene | 4.63 | 4.70 |
| Methanol | 4.26 | 3.99 |
| Monoterpenes | 0.86 | 0.38 |
| Total | 9.75 | 9.07 |

Accordingly, HCHO yield from BVOCs emission was estimated to be about $9.07 \times 10^9$ g, and mostly produced from isoprene and methanol. The calculated HCHO yield from BVOCs emission is similar to that of previous study during 2005-2016 (Shen et al., 2019). In addition, compared with anthropogenic sources, HCHO yield from BVOCs is much smaller, which indicates that the anthropogenic is the main contributor of secondary production of HCHO in Shanghai (Shen et al., 2019; Fan et al., 2021).

As shown in Table R4, we have also reviewed the related studies about the BVOCs emissions in Shanghai and its surrounding areas (the Yangtze River Delta) in relevant years. Wang et al. (2021a) assessed the impacts of land cover change and climate variability on BVOCs emissions in China from 2001 to 2016, in which variations of BVOCs emissions in Shanghai over the years were extremely small. Considering the different input dataset and settings would bring large differences in the simulated results, it was unfeasible to use BVOCs emissions from different studies for the investigation of temporal variation. Therefore, the BVOCs emissions in 2014 were used to basically characterize the approximate level of BVOCs from 2010 to 2016 in this study.

*Table R4. Comparison of simulated BVOCs emissions in Shanghai (SH) and the Yangtze River Delta (YRD) based on MEGAN.*

| Simulated year | Reference | MEGAN version | Region | BVOCs emission ($10^4$ t) |
|---|---|---|---|---|
| 2010 | Song et al. (2012)[1] | V 2.04 | YRD | 110 |
|  |  |  | SH | 0.122 |
| 2014 | Liu et al. (2018a; 2018b)[2] | V 2.10 | YRD | 188.6 |
|  |  |  | SH | 1.2 |
| 2016 | Wang et al. (2021b)[3] | V 3.1 | YRD | 162 [1] |
|  |  |  | SH | ~ 0.34 [4] |

*[1] Total annual emission inferred from the simulated BVOCs emissions in January, April, July and October.*

*[2] A variety of methods were used to reduce the uncertainty of plant functional types (PFT) database. The proportions of dominant components of BVOCs were also provided.*

*[3] BVOCs emission was simulated without drought stress.*

*[4] It is BVOCs emissions in July, which has been inferred from Fig. S3 of Wang et al. (2021b).*

In addition, it should be noted that HCHO yield was also impacted by the $NO_x$ levels, e.g. $RO_2$ radical from VOCs react with $HO_2$ to from organic peroxides under low $NO_x$ condition. This process reduces the reaction of $RO_2$ and NO, which in turn decreases the production of HCHO, therefore, HCHO yield from VOCs is proportional to $NO_x$ condition (Palmer et al., 2006; Marais et al., 2012; Miller et al., 2017). In this study, the estimation using a fixed HCHO yield may overestimate HCHO production in later years due to the decreases of $NO_x$ in Shanghai (Xue et al., 2020). In previous studies, the proportional relationship between HCHO yield and $NO_x$ condition was usually obtained when 1 ppbv of $NO_x$ regard as the high condition, and 0.1 ppbv of $NO_x$ regard as low condition (Palmer et al., 2006; Marais et al., 2012; Miller et al., 2017). However, the $NO_x$ concentration in Shanghai is still relatively higher (basically 30-60 ppbv in urban) (Gao et al., 2017). Therefore, in such a high $NO_x$ condition, the effect of $NO_x$ decreases on HCHO yield needs to be further studied.

Minor comments:
1. Page 2, line 39-40, please include GOME-2 A, B, and C
R: Thanks for the suggestion. We have added GOME-2 A, B, and C in the Introduction, please refer to Line 39.

2. Page 3, line 76, please change "in a day" to "in the daytime"
R: We have corrected 'in a day' to 'in the daytime'. Please refer to Line 76.

3. Page 3, line 80, please change "pixels" to "rows"
R: We have corrected 'pixels' to 'rows'. Please refer to Line 80.

4. Page 3, line 81, here and elsewhere, please change "~" to "-". "~" means "approximately"
R: Thanks for the reminding. We have changed '~' to '-'. Please refer to Line 81.

5.  Page 3, line 92-93, I think by using MainQualityFlag=0 as the criterion, you have filtered out pixels affected by row anomalies already.

R: Thanks for your comments. Pixels flagged with 0 in field MainDataQualityFlag were considered to be passed quality check. However, in OMHCHO V3.0 product, field XtrackQualityFlags has been carried over from the L1b product to characterize pixels affected by the row anomaly. We have tested and found that the usage of XtrackQualityFlags during the filtering process can affect the results. As shown in Figure R3, under the two filtering conditions, HCHO VCDs have obvious differences in some regions. In addition, Figure R3(a) exists a track with abnormally high value, which basically disappeared after being filtered by using XtrackQualityFlags. It means that including XtrackQualityFlags in filtering criterion in addition to MainDataQualityFlag would be effective.

[Figure]

*Figure R3. Spatial distribution of HCHO VCDs of Shanghai and surrounding areas in October 2010 under different filter conditions of (a) without XtrackQualityFlags and (b) with XtrackQualityFlags.*

6.  Page 5, line 127-129, could you please explain the spatial variations in HCHO columns. Is the spatial pattern consistent with emissions?

R: Thanks for your comments. For determining whether the spatial variations in HCHO VCD is consistent with local emissions, we have displayed the spatial distribution of anthropogenic NMVOCs in Shanghai in 2017 with high resolution of 4 km×4 km in Figure R4 (An et al., 2021). It can be seen that the hotspots of NMVOCs emission were concentrated in the city center, while the highest HCHO VCD appeared in the relatively remote Qingpu district, where the NMVOCs emissions were relatively low. Obviously, the anthropogenic emissions and HCHO VCD do not coincide well in the spatial pattern, and the high value of HCHO VCD in Qingpu district cannot be directly explained by the emission of anthropogenic NMVOCs.

Not only high HCHO VCD observed by satellites, but also high concentrations of surface HCHO have been measured in Qingpu District (e.g. Su et al., 2019; Zhang et al., 2021). Impact of air masses transport containing high concentrations of reactive VOCs from adjacent Zhejiang Province and Jiangsu Province was reported (Zhang et al., 2020). As shown in Figure R4, the anthropogenic NMVOCs inventory in the Yangtze River Delta also displays that there were obvious sources of NMVOCs in the

southern part of Jiangsu and the northern part of Zhejiang. The high local atmospheric oxidation capacity leads to the rapid degradation of VOCs, which in turn enhances the contributions of anthropogenic NMVOCs to the local HCHO production in Qingpu district (Zhang et al., 2021).

In addition, model simulation showed that isoprene plays an important role in the production of HCHO as the precursor, and it can contribute about 36% of the production of HCHO during $O_3$ formation episodes from April to June in 2018 in Qingpu district (Zhang et al., 2021). The abundant BVOCs in Qingpu district may be another important reason for the high HCHO VCD. Please refer to Line 139-143.

[Figure]

*Figure R4. Anthropogenic NMVOCs emissions in Shanghai in 2017. The inventory data is available from An et al. (2021).*

**Reference:**

An, J. Y., Huang, Y. W., Huang, C., Wang, X., Yan, R. S., Wang, Q., Wang, H. L., Jing, S. A., Zhang, Y., Liu, Y. M., Chen, Y., Xu, C., Qiao, L. P., Zhou, M., Zhu, S. H., Hu, Q. Y., Lu, J., and Chen, C. H.: Emission inventory of air pollutants and chemical speciation for specific anthropogenic sources based on local measurements in the Yangtze River Delta region, China, Atmos Chem Phys, 21, 2003-2025, https://doi.org/10.5194/acp-21-2003-2021, 2021.

Fan, J. C., Ju, T. Z., Wang, Q. H., Gao, H. Y., Huang, R. R., and Duan, J. L.: Spatiotemporal variations and potential sources of tropospheric formaldehyde over eastern China based on OMI satellite data, Atmos Pollut Res, 12, 272-285, https://doi.org/10.1016/j.apr.2020.09.011, 2021.

Gao, W., Tie, X. X., Xu, J. M., Huang, R. J., Mao, X. Q., Zhou, G. Q., and Chang, L. Y.: Long-term trend of O-3 in a mega City (Shanghai), China: Characteristics, causes, and interactions with precursors, Sci Total Environ, 603, 425-433, https://doi.org/10.1016/j.scitotenv.2017.06.099, 2017.

Liu, Y., Li, L., An, J., Zhang, W., Yan, R., Huang, L., Huang, C., Wang, H., Wang, Q., and Wang, M.:

Emissions, Chemical Composition, and Spatial and Temporal Allocation of the BVOCs in the Yangtze River Delta Region in 2014, ENVIRONMENTAL SCIENCE, 39, 608-617, 2018a.

Liu, Y., Li, L., An, J. Y., Huang, L., Yan, R. S., Huang, C., Wang, H. L., Wang, Q., Wang, M., and Zhang, W.: Estimation of biogenic VOC emissions and its impact on ozone formation over the Yangtze River Delta region, China, Atmos Environ, 186, 113-128, https://doi.org/10.1016/j.atmosenv.2018.05.027, 2018b.

Marais, E. A., Jacob, D. J., Kurosu, T. P., Chance, K., Murphy, J. G., Reeves, C., Mills, G., Casadio, S., Millet, D. B., Barkley, M. P., Paulot, F., and Mao, J.: Isoprene emissions in Africa inferred from OMI observations of formaldehyde columns, Atmos Chem Phys, 12, 6219-6235, https://doi.org/10.5194/acp-12-6219-2012, 2012.

Miller, C. C., Jacob, D. J., Marais, E. A., Yu, K. R., Travis, K. R., Kim, P. S., Fisher, J. A., Zhu, L., Wolfe, G. M., Hanisco, T. F., Keutsch, F. N., Kaiser, J., Min, K. E., Brown, S. S., Washenfelder, R. A., Abad, G. G., and Chance, K.: Glyoxal yield from isoprene oxidation and relation to formaldehyde: chemical mechanism, constraints from SENEX aircraft observations, and interpretation of OMI satellite data, Atmos Chem Phys, 17, 8725-8738, https://doi.org/10.5194/acp-17-8725-2017, 2017.

Palmer, P. I., Abbot, D. S., Fu, T. M., Jacob, D. J., Chance, K., Kurosu, T. P., Guenther, A., Wiedinmyer, C., Stanton, J. C., Pilling, M. J., Pressley, S. N., Lamb, B., and Sumner, A. L.: Quantifying the seasonal and interannual variability of North American isoprene emissions using satellite observations of the formaldehyde column, J. Geophys. Res.-Atmos., 111, 14, https://doi.org/10.1029/2005jd006689, 2006.

Shen, L., Jacob, D. J., Zhu, L., Zhang, Q., Zheng, B., Sulprizio, M. P., Li, K., De Smedt, I., Abad, G. G., Cao, H. S., Fu, T. M., and Liao, H.: The 2005-2016 Trends of Formaldehyde Columns Over China Observed by Satellites: Increasing Anthropogenic Emissions of Volatile Organic Compounds and Decreasing Agricultural Fire Emissions, Geophys Res Lett, 46, 4468-4475, https://doi.org/10.1029/2019gl082172, 2019.

Song, Y., Zhang, Y., Wang, Q., and An, J.: Estimation of biogenic VOCs emissions in Eastern China based on remote sensing data, Acta Sci Circum, 32, 2216-2227, 2012.

Su, W. J., Liu, C., Hu, Q. H., Zhao, S. H., Sun, Y. W., Wang, W., Zhu, Y. Z., Liu, J. G., and Kim, J.: Primary and secondary sources of ambient formaldehyde in the Yangtze River Delta based on Ozone Mapping and Profiler Suite (OMPS) observations, Atmos Chem Phys, 19, 6717-6736, https://doi.org/10.5194/acp-19-6717-2019, 2019.

Wang, H., Wu, Q. Z., Guenther, A. B., Yang, X. C., Wang, L. N., Xiao, T., Li, J., Feng, J. M., Xu, Q., and Cheng, H. Q.: A long-term estimation of biogenic volatile organic compound (BVOC) emission in China from 2001-2016: the roles of land cover change and climate variability, Atmos Chem Phys, 21, 4825-4848, https://doi.org/10.5194/acp-21-4825-2021, 2021a.

Wang, Y. J., Tan, X. J., Huang, L., Wang, Q., Li, H. L., Zhang, H. Y., Zhang, K., Liu, Z. Y., Traore, D., Yaluk, E., Fu, J. S., and Li, L.: The impact of biogenic emissions on ozone formation in the Yangtze River Delta region based on MEGANv3.1, Air Qual. Atmos. Health, 14, 763-774, https://doi.org/10.1007/s11869-021-00977-0, 2021b.

Xue, R. B., Wang, S. S., Li, D. R., Zou, Z., Chan, K. L., Valks, P., Saiz-Lopez, A., and Zhou, B.: Spatio-temporal variations in $NO_2$ and $SO_2$ over Shanghai and Chongming Eco-Island measured by Ozone Monitoring Instrument (OMI) during 2008-2017, J Clean Prod, 258, 14, https://doi.org/10.1016/j.jclepro.2020.120563, 2020.

Zhang, K., Li, L., Huang, L., Wang, Y. J., Huo, J. T., Duan, Y. S., Wang, Y. H., and Fu, Q. Y.: The impact of volatile organic compounds on ozone formation in the suburban area of Shanghai, Atmos Environ,

232, 11, https://doi.org/10.1016/j.atmosenv.2020.117511, 2020.

Zhang, K., Huang, L., Li, Q., Huo, J. T., Duan, Y. S., Wang, Y. H., Yaluk, E., Wang, Y. J., Fu, Q. Y., and Li, L.: Explicit modeling of isoprene chemical processing in polluted air masses in suburban areas of the Yangtze River Delta region: radical cycling and formation of ozone and formaldehyde, Atmos Chem Phys, 21, 5905-5917, https://doi.org/10.5194/acp-21-5905-2021, 2021.

Zhu, L., Jacob, D. J., Mickley, L. J., Marais, E. A., Cohan, D. S., Yoshida, Y., Duncan, B. N., Abad, G. G., and Chance, K. V.: Anthropogenic emissions of highly reactive volatile organic compounds in eastern Texas inferred from oversampling of satellite (OMI) measurements of HCHO columns, Environ Res Lett, 9, 7, https://doi.org/10.1088/1748-9326/9/11/114004, 2014.

Zhu, L., Mickley, L. J., Jacob, D. J., Marais, E. A., Sheng, J. X., Hu, L., Abad, G. G., and Chance, K.: Long-term (2005-2014) trends in formaldehyde (HCHO) columns across North America as seen by the OMI satellite instrument: Evidence of changing emissions of volatile organic compounds, Geophys Res Lett, 44, 7079-7086, https://doi.org/10.1002/2017gl073859, 2017.

---

## Author Response (AR2)

**Response to reviewer**

We thank the reviewer for the constructive comments and suggestions, which are very helpful for the improvement of the manuscript. We have revised the manuscript appropriately and addressed all the reviewer's comments point-by-point for consideration as below. The remarks from the reviewer are shown in black, and our responses are shown in blue color. All the page and line numbers mentioned following are refer to the revised manuscript without change tracked.

The authors have added some new analysis to answer the reviewers' questions, though the overall results seem to be the same as the previous manuscript. There are some additional issues with data processing and HCHO yield calculation, which I think should be addressed carefully before being acceptable for publication:

1. The authors mention the strip patterns are caused by the data processing method they use (Method 2). Using Method 1 could largely remove the strip patterns. It's unclear to me why the authors insist using Method 2 to process data, while Method 1 is clearly much better. I'd suggest the authors use Method 1, otherwise readers will likely interpret the artificial patterns shown in Figure 2 as errors with satellite retrievals.

R: Thanks for the comments. Due to the employment of smoothing procedure in Method 1, the strip patterns were largely removed. We agreed that this kind of strip patterns could be misunderstood as the errors with satellite retrievals. So we finally decided to follow the reviewer's suggestion using Method 1 for the data processing. Except for the strip pattern in spatial distribution, the annual, monthly or seasonal averages for the Shanghai area changed very slight. The relevant figures and reported values were all replaced with the updated one, including Figure 1, 2, 3, 4, 5, 6, 7, 9, 10, Figure S1, and Figure S2. Please refer to the revised manuscript and supplementary.

Moreover, we also want to clarify that the strip patterns were not caused by the Method 2 itself previously, but considered to be inherited from the satellite product without the smoothing procedure.

2. Table R2 is confusing to me. The authors show the HCHO yield from secondary VOCs is higher than primary emission by nearly two orders of magnitude, which doesn't sound right to me. How did you calculate HCHO yield? What is the speciation of VOCs for each sector? More details are needed here.

R: Thanks for the constructive comments. Table R2 in last responses document (RC1) summarized the estimated May-September total HCHO production in eastern China in 2010 from CB05 and SAPRC07 mechanism species, and the comparison with the reference (Shen et al., 2019). We're not sure whether the reviewer want to pointed to Table R1, which listed the primary emissions and estimated secondary productions of HCHO from anthropogenic NMVOCs based on SAPRC07 mechanism species, and showed the secondary production is higher than primary emission by nearly two orders of magnitude.

Here we have replied this concern from three aspects.

(1) Calculation of HCHO yield

Taking year of 2010 as an example, we have detailed described the calculation process of secondary HCHO production as following. The total anthropogenic NMVOCs emission of Shanghai in MEIC is about $58.21 \times 10^4$ t, of which industrial, power, residential and transportation is $44.52 \times 10^4$, $0.15 \times 10^4$, $2.61 \times 10^4$ and $10.93 \times 10^4$ t, respectively.

Based on the MEIC methodology, the total NMVOCs emission can be split into individual species and then mapped into various chemical mechanisms that are configured in the chemical transport models, such as SAPRC 07, CB05, RACM and GEOS-Chem, etc. (Li et al., 2014). In this study, the total NMVOCs have been speciated into 36 species individually for each sector using SAPRC07 mechanism. Figure R1 shows the speciation of NMVOCs for each sector. For example, there were $5.81 \times 10^9$ mol NMVOCs species in total for industry sector, respective to the total emission of $4.452 \times 10^{11}$ g as mentioned above, the averaged molecular weight for the speciated NMVOCs is about 76.6 g mol$^{-1}$. So far, no obvious errors in the NMVOCs speciation were found.

[Figure]

*Figure R1. The speciation and emissions of NMVOCs of Shanghai for each sector in 2010.*

Then, we can determine the HCHO yield of each species of each sector individually, and summed up to get the secondary HCHO from anthropogenic NMVOCs for a given sector, as expressed in Eq. (R1).

$$S = M_{HCHO} \sum_i E_i n_i Y_i \qquad (R1)$$

where S (unit: g) is the secondary HCHO production from anthropogenic NMVOCs for a given sector, $M_{HCHO}$ (unit: g mol$^{-1}$) represents the molecular weight of formaldehyde. For each sector, $E_i$ (unit: mol) is the primary emission of NMVOC speciation $i$, as shown in Figure R1. $n_i$ stands for the number of carbons in the species $i$. $Y_i$ is the HCHO yield per carbon for species $i$, as referred to the 1-day HCHO yields under high-NOx conditions from VOCs in Shen et al. (2019).

In order to explain clearly, we used the species emission of acetaldehyde of transportation sector to show the calculation process. In Fig. R1(d), the transportation emitted primarily acetaldehyde (indicated as CCHO in SAPRC07) was $33.91 \times 10^6$ mol in 2010. According to the corresponding HCHO yield per carbon ($Y_{CCHO}$=0.5) and number of carbons in acetaldehyde ($n_{CCHO}$=2), the secondary HCHO production from acetaldehyde in transportation was calculated to be $1.02 \times 10^9$ g. Similarly, all the species for each sector can be further calculated and summed up, as listed in Table S1.

It's important to pointed that the emitted NMVOC species are impossible to be oxidized totally to HCHO, and the calculated secondary production from anthropogenic NMVOC should be overestimated significantly than they actually produced HCHO in the real atmosphere. This results just indicate the potential of anthropogenic secondary production of HCHO, and do not represent the accurate production in the environment. We have also clarified it in the revised manuscript, and please refer to Line 193-194.

2) Quantification of primary emission

As mentioned above, the primary HCHO emission of each sector can be obtained during the speciation procedure, marked with red rectangle in Fig. R1. Meanwhile, we double checked the primary anthropogenic HCHO emission by comparing with some previous literatures, shown in Figure R2. It can prove that the primary anthropogenic HCHO emission only occupied a small proportion in total anthropogenic NMVOC emission in Shanghai (1-2%), Beijing (~3%) and surrounding area (~2%), China (~2%) and the Asia (1-2%). Combined with the total anthropogenic NMVOCs emission of Shanghai, we considered the estimated primary anthropogenic HCHO emission as a reasonable amount.

[Figure]

*Figure R2. Comparisons of proportion of primary anthropogenic HCHO emissions to total anthropogenic NMVOCs emissions with other studies.*

3) Explanations for the huge difference between primary emission and secondary production from anthropogenic NMVOCs.
The above two aspects showed that the primary emission and secondary production of HCHO from anthropogenic NMVOCs were estimated in a reasonable way. But the huge difference between primary emission and secondary production is largely deviated from the common sense as the reviewer suggested.

We think that the amount of primary HCHO emission from anthropogenic is quite reliable compared the overestimated secondary production. In the calculation of secondary production from anthropogenic NMVOC, we used the HCHO yield from Shen et al. (2019), which is computed for 1-day HCHO yields under high-$NO_x$ conditions from VOCs and comparable with other studies (Chan Miller et al., 2016; Zhu et al., 2014). However, not all the primary anthropogenic NMVOCs are engaged in the reaction to form HCHO, this calculated secondary production just stands for the potential, not the real production or contribution.

In addition, we have to stated that previous studies of HCHO attempting to unravel the role of secondary production versus direct emission vary widely in their estimates and the conclusions are varied in spatial and temporal (e.g. Luecken et al., 2018; Stroud et al., 2016; Ling et al., 2017; Liu et al., 2020). In addition, even for the same area but different techniques, the secondary production of total HCHO were determined from 30-50% to 95% (and mostly point sources 92%) (Zhang et al., 2013; Parrish et al., 2012). Therefore, the exact conclusion about the ratio of HCHO yielded from secondary production of VOCs to primary emission is difficult to drawn from the existing discussions in this study, and the possibility of such magnitude differences can not be excluded completely neither.

3. The authors show secondary production of HCHO is over 30 times higher for anthropogenic emissions (300 * 10^9 g) than biogenic emissions (9 * 10^9 g). This shouldn't be right. We know that globally biogenic emissions are the largest contributor to HCHO. Even for urban areas, the contribution from anthropogenic vs. biogenic emissions should be comparable. I'd suggest the authors double check their methods for calculating HCHO yield, which may also explain my previous question.

R: Thanks for the constructive comments. In view of the HCHO columns, they are commonly found in large quantities over forested regions in the world, however, a large fraction of HCHO columns was observed in industrial/urban regions in East Asia, that HCHO levels were 19-33% higher in urban regions than forested regions (Souri et al., 2017). While the biogenic emissions are contributed significantly to HCHO in global scale, large values of HCHO observed in megacities indicate the predominance of anthropogenic sources.

Figure R3 displays the land surface type of Shanghai, which shows the high degree of urbanization and less coverage of broadleaf trees. Broadleaf trees are considered the biggest contributor to isoprene, while crops are weak isoprene emitters (Stavrakou et al., 2014; Li et al., 2016). We have also added Figure R3 as Figure S3 in the supplementary. Therefore, in such an urbanized and densely populated megacity Shanghai, the biogenic VOC emissions are expected to be much minor compared to forested regions (Liu et al., 2018; Wu et al., 2020a; Li et al., 2020). The biogenic VOCs emissions about $1.2 \times 10^4$ t of Shanghai in 2014, used to infer the HCHO yields, is comparable with the reported values in those literatures. Therefore, we would not expect more HCHO yield from biogenic emission than current estimation.

[Figure]

*Figure R3. The spatial distribution of surface land types in Shanghai (data from ESA Land Cover Product, available at: http://maps.elie.ucl.ac.be/CCI/viewer/).*

So we think the contribution to HCHO of biogenic emissions may not be comparable to anthropogenic sources in Shanghai. In Beijing-Tianjin-Heibei area of China, the

anthropogenic VOCs might impose substantial impacts more, especially in urbanized areas (Zhu et al., 2018). Based on the two models (the Positive Matrix Factorization (PMF) model and a photochemical box model with Master Chemical Mechanism (PBM-MCM)), the contribution of isoprene, indicating the biogenic emission, to secondary HCHO formation was about 9.6± 3% in July 2006 at an urban site in the Pearl River Delta (PRD), China, much lower than that at semi-rural site in Guangzhou about 43.2% (Ling et al., 2017). It suggested the contribution of biogenic emissions to secondary HCHO production in urban site may not be important as much as expected like other places, even in summer season.

Finally, we would like to conclude that the secondary production plays a dominant role for HCHO compared to direct emission, however, the relative importance of anthropogenic or biogenic emission contributed to secondary production are still in disputed, which is strongly associated to the time and location, and the results also will be impacted by the adopted methodology (e.g. Wang et al., 2017; Bastien et al., 2019; Wu et al., 2020b). The quantifications of specific anthropogenic source types or VOC precursors contributing to HCHO have less been done so far, need to be further investigated.

Back to the manuscript context, we used the primary emission of HCHO from MEIC in the initial submission, in order to justify the impacts of anthropogenic directly emissions. As the reviewer suggested, we should not overlook the important secondary production. So we have estimated roughly the secondary production from anthropogenic and biogenic emissions. This is a very insightful comments and we do want to take more efforts on this issue which are not well explained by others. But considering the main focus of the manuscript, we would appreciate reviewer's understanding that we would not discuss too much regarding whether the secondary production from biogenic or anthropogenic contribute more. We hope this manuscript can prompt the perspective on the inferring ozone sensitivity from satellite $HCHO/NO_2$ ratio revised with ground surface measurement.

[revised manuscript text omitted]